# Flux coupling approach on an exchange grid for the IOW Earth System Model (version 1.04.00) of the Baltic Sea region

Sven Karsten[1], Hagen Radtke[1], Matthias Gröger[1], Ha T. M. Ho-Hagemann[2], Hossein Mashayekh[1], Thomas Neumann[1], and H. E. Markus Meier[1]

[1]Leibniz Institute for Baltic Sea Research Warnemünde (IOW), Seestraße 15, 18119 Rostock, Germany
[2]Helmholtz-Zentrum Hereon, Max-Planck-Straße 1, 21502 Geesthacht, Germany

**Correspondence:** Sven Karsten (sven.karsten@io-warnemuende.de)

**Abstract.** In this article the development of a high-resolution Earth System Model (ESM) for the Baltic Sea region is described. In contrast to conventional coupling approaches, the presented model features an additional (technical) component, the *flux calculator*, that calculates fluxes between the model components on a common *exchange grid*. This approach naturally ensures conservation of exchanged quantities, a locally consistent treatment of the fluxes and facilitates interchanging model components in a straightforward manner. The main purpose of this model is to downscale global reanalysis or climate model data to the Baltic Sea region since typically global model grids are too coarse to resolve the region of interest sufficiently. The regional ESM consists of the Modular Ocean Model 5 (MOM5) for the ocean and the COSMO model in CLimate Mode (CCLM, version 5.0_clm3) for the atmosphere. The bi-directional ocean-atmosphere coupling allows for a realistic air-sea feedback which outperforms the traditional approach of using uncoupled standalone models as typically pursued with the EURO-CORDEX protocol. In order to address marine environmental problems (e.g. eutrophication and oxygen depletion), the ocean model is internally coupled to the marine biogeochemistry model ERGOM set up for the Baltic Sea's hydrographic conditions. The regional ESM can be used for various scientific questions such as climate sensitivity experiments, reconstruction of ocean dynamics, study of past climates and natural variability as well as investigation of ocean-atmosphere interactions. Therefore, it can serve for better understanding of natural processes via attribution experiments that relate observed changes to mechanistic causes.

## 1 Introduction

The European continent and its marginal seas are located between the polar climate zone in the north and subtropical climate in the south and are likewise influenced by temperate maritime climate in the west and continental climate with high seasonal amplitudes in the east. Consequently, the climate of Europe is highly variable, resulting in many different climate zones to be distinguished (Köppen and Geiger, 1930). These circumstances make this region a challenge for the development of coupled ESMs (Gröger et al., 2021). This is in particular the case for the Baltic Sea region which is known for its high natural variability, complicated coast lines given by numerous islands, narrow channels between the basins and the small baroclinic Rossby radius (Fennel et al., 1991) resulting from a permanent haline stratification.

Thus, simulating the Baltic Sea's regional climate requires a sufficiently high spatial resolution of the oceanic model grid.
However, the corresponding atmospheric circulation is usually simulated on a much larger domain, since the pathways of cyclones originating from the North Atlantic region should be part of it. For this reason, the atmospheric model cannot be discretized with the same high resolution as the ocean model at reasonable numerical costs. Hence, a more adequate strategy is needed to provide the highly resolved oceanic information to a suitable atmospheric model simulating the Baltic Sea's regional climate.

For the recent past, there are appropriate measurement data (or derived products such as satellite data) for the Baltic Sea's surface variables available that may serve as the lower boundary for the atmospheric model. This enables uncoupled simulations where the atmosphere is simulated first, using observed values for the ocean state, and then an ocean model can later be driven with the atmospheric variables as forcing. This strategy, however, naturally fails for future projections. Projections for the Earth's future climate are based on global ESMs, i.e. platforms that interactively couple different components of the earth system (e.g. atmosphere, biosphere, cryosphere, ocean, e.g. (Heinze et al., 2019)). Still, the resolution of these models is insufficient to explicitly resolve important small scale processes (e.g. land-sea-mask effects, polar lows, etc.) leading to e.g. unrealistic wind fields over the Baltic Sea (Meier et al., 2011). Therefore, regional models were developed which represent a step forward to more sophistically include small scale processes and more realistically represent orography. However, the majority of these models for the Northern European region consists only of a single standalone model for the atmosphere that is driven by input data either from global models or reanalysis products at the model boundaries. For future projections, this approach is problematic as input information can only be derived from global models which can have substantial biases in the region of interest (especially at coastal regions, where the coarse resolution of the land-sea mask can become insufficient). While numerous tested methods exist to bias-correct model forcing data for the historical period (e.g. (Teutschbein and Seibert, 2012; Vaittinada Ayar et al., 2021)) their application for future periods can not be validated and may be therefore problematic. In addition, these models employ bulk formulas for the exchanged fluxes with rather simplistic models for the transfer coefficients. This argues for the development of fully coupled Regional Earth System Model (RESMs) for future projections that consistently account for the local peculiarities of the considered domain (Gröger et al., 2021).

For the Baltic Sea, two independent coupled model systems from the Danish Meteorological Institute and the Swedish Meteorological and Hydrological Institute demonstrated an improvement of simulated winter Sea Surface Temperatures (SSTs) compared to their corresponding ocean only simulations (Tian et al., 2013; Gröger et al., 2015). However, the fluxes are calculated entirely by the atmospheric model on its coarser grid.

In contrast to the existing RESMs for the Baltic Sea region, the IOW ESM presented here involves a third component, called the flux calculator, that computes the fluxes on an exchange grid formed by the intersections between the two model grids. This approach naturally ensures a locally consistent treatment of the fluxes and conservation of exchanged quantities. Thus, the drawbacks coming with the high resolution of the oceanic model grid and the large atmospheric simulation domain can be circumvented. Moreover, this approach enables more flexibility in interchanging model components and thus simplifies the development. At the present stage the IOW ESM consists of the MOM5 model (Neumann et al., 2021) for the Baltic Sea and the CCLM model (version 5.0_clm3) (Steger and Bucchignani, 2020) for the atmosphere on the EURO-CORDEX domain.

The exchange grid method is not new but was introduced by Balaji et al. (2006) for the Flexible Modeling System of the Geophysical Fluid Dynamics Laboratory at the Princeton University. Also the coupler Earth System Modelling Framework with the National Unified Operational Prediction (ESMF/NUOPC) follows this philosophy, where a *mediator component* can be used as an equivalent to our flux calculator, and the exchange grid calculation is performed on-the-fly during the model runtime (Campbell and Whitcomb, 2013). From the global modeling perspective the National Center of Atmospheric Research (NCAR) Community Earth System Model (CESM) (Danabasoglu et al., 2020) follows a similar strategy of consistently calculating fluxes by an additional component corresponding to the presented flux calculator. An alternative approach to a conservative mapping was introduced by Furevik et al. (2003), where the action of an exchange grid was mimicked by a stochastic sampling in the fashion of a Monte Carlo simulation. However, for the Baltic Sea area, our model system is, to our best knowledge, the first to fully employ this approach. The ICONGETM coupled model system (Bauer et al., 2021) used an ESMF/NUOPC exchange grid before, but only for a conservative mapping of fluxes, the flux calculations were still performed by the atmospheric model component on its own grid, not by the mediator.

The manuscript is structured as follows. First, the theoretical background and the methodology of the developed coupling approach is described in Sect. 2 including implementation details. In order to investigate the differences between the chosen exchange grid and the more traditional coupling strategies, reference runs for different types of exchange grids with ERA5 (Hersbach et al., 2019) reanalysis data as atmospheric boundaries have been performed and are compared and discussed in Sect. 3. The focus of this article is on the flexibility in model development, the consistency of the presented flux calculation and the facilitation of running simulations as well as performing subsequent data analysis that is enabled by the presented framework. Still, the presented simulations are performed with realistic setups for the model components for several decades and thus give a robust impression on the model performance. Finally, the work is concluded in Sect. 4 and further details can be found in the Appendix.

## 2 Methods

One basic problem when dealing with RESMs is that the individual components (atmosphere, ocean, land, etc.) are described by different models that act on different grids, see Fig. 1 and Fig. 2.

Still, the components have to be coupled in order to communicate their state to each other and exchange fluxes as in reality.

### 2.1 The exchange grid and the flux calculator

In the standard approach to couple climate model components, the exchanged fluxes are calculated in the atmospheric model (Wang et al., 2015; Sein et al., 2015). Since fluxes naturally depend on the state of the ocean, the corresponding information has to be communicated to the atmosphere first. Due to the normally lower resolution of the atmospheric grid, the ocean's state information has to be averaged (weighted by areas) over several ocean grid cells and typically over different surface types (water, different ice classes or land), see blue and white boxes in Fig. 2 and for more details Fig. B1 in Appendix B.

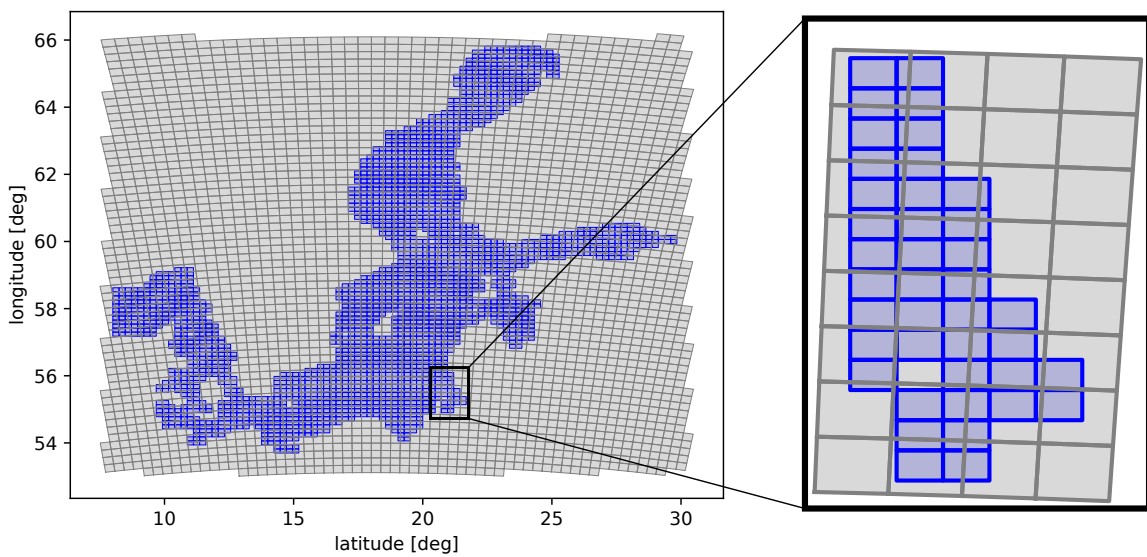

**Figure 1.** Overlaying grids of atmospheric and ocean models for the Baltic Sea.

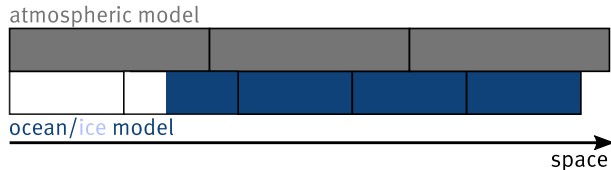

**Figure 2.** Schematic of atmosphere and ocean model grids. The bottom model can support different surface types as for instance water (blue) and ice (white). However, both are typically represented by the same grid cell and only the concentration of each surface type in that grid cell is considered. In the illustrated case, the first left ocean model grid cell has an ice concentration of 100% and the second left has roughly 40% ice and 60% liquid water, whereas all other cells are fully covered with water.

With the averaged state information from the ocean model (ice or land model) and its own internal state, the flux can be calculated (as a field on the atmospheric grid) by the atmospheric model. Subsequently, the flux field has to be redistributed on the ocean (sea-ice or land) cells (again in a area-weighted manner such that the exchanged quantity is overall conserved, i.e. a conservative mapping), see Fig. B1 panel (b). Since the flux is only calculated from averaged information, this approach is locally not consistent and can become inaccurate. This is especially true if many bottom grid cells are covered by one
atmospheric grid cell.

  The alternative approach chosen within the developed ESM is the introduction of a third component, i.e. the flux calculator that acts on an exchange grid. The most natural choice for such an exchange grid would be the set of intersections between the

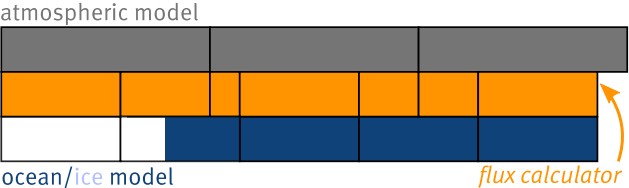

**Figure 3.** Introduction of the exchange grid (orange boxes) on which the *flux calculator* is acting.

atmospheric and the ocean grid cells. This grid has, by construction, a higher resolution than all involved model components (see Fig. 3). Thus, the exchange grid is capable to resolve *all* peculiarities covered by the involved model grids.

Employing the aforementioned exchange grid, the example from above, i.e. fluxes shall be communicated between the atmosphere and the ocean, is then treated as follows. First, the model components of the coupled model send their necessary state variables to the flux calculator. The variables are thereby mapped onto the exchange grid, see Fig. 4 panel (a), via conservative mapping. Importantly, since the intersection exchange-grid cells are always smaller or equal to the grid cells of the models, this mapping does not feature any averaging and, thus, no information is lost. Moreover, different surface types can be treated
individually since this information on features of the ocean, sea-ice or land model can be implemented in the flux calculator.

     Second, with all the state information, the flux calculator is then able to calculate the flux of interest. Any formula can be used that derives the desired fluxes from the available state variables. The calculation only requires local information and can be surface-type-dependent. The resulting flux has to be finally mapped onto the bottom grid, see Fig. 4 panel (b), again via conservative mapping. Note that, although not shown in the figures (for the sake of clarity), the exactly same fluxes are
communicated to the atmospheric model as well. This ensures a conservative and locally consistent exchange of mass, energy and momentum between the different model components.

     However, the calculation of some fluxes does not depend only on surface fields and therefore are out of scope of the flux calculator capabilities. In particular, precipitation and (downward shortwave and longwave) radiative fluxes are calculated entirely by the atmospheric model and the resulting fluxes are sent via flux calculator to the ocean model. Nevertheless, there is
no direct communication between the two model components and this simplifies ultimately interchangeability of the models. This is due to the fact, that either model can be exchanged (in principle) without affecting the source code of the other; only the self-developed flux calculator module has to be adapted.

     In order to investigate the impact of the described exchange grid approach, two alternative exchange grid types are considered for comparison.

**2.2   Different exchange grids**

In Sect. 2.1, the described exchange grid is formed from the intersections of the involved models grids, henceforth called the *intersection grid*. The two apparent alternative exchange grids can then be either the *atmospheric model grid* or the *ocean model grid* itself.

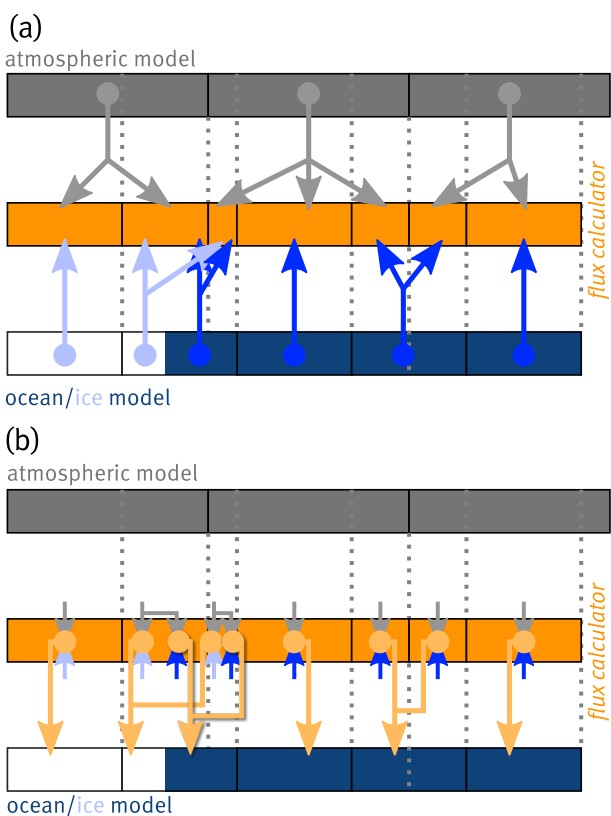

**Figure 4.** Coupling the models via the exchange grid and the flux calculator. Panel (a): State variables, calculated in the respective models (marked by the filled circles), are communicated to the flux calculator (as visualized by the arrows) without averaging. Panel (b): Fluxes are calculated on the exchange grid and subsequently communicated to the bottom model.

Since for a typical coupled model setup, the atmospheric grid has the lower resolution than the ocean model, the two resulting
alternative exchange grids can differ quite substantially from each other as well as from the intersection grid. In any case both alternatives will have (by construction) a lower resolution than the intersection-type exchange grid. With this more general conception of an exchange grid we are now able to consider three different kinds of coupling approaches on equal footing.

First, we consider the approach introduced in Sect. 2.1 and calculate fluxes by the flux calculator with state variables locally resolved on the intersection grid and subsequently communicate the fluxes to the models. Second and third, we may employ
each of the model grids as the exchange grid and calculate fluxes with spatially averaged fields and communicate then the fluxes to the involved models. These two last cases include also the typical coupling approach, i.e. using a conservative mapping of state variables from the ocean to the atmospheric model accompanied by the flux calculation via the latter and the communication back to the former (e.g. (Wang et al., 2015)).

Importantly, the developed flux calculator methodology enables to investigate all three approaches with the same infrastruc-
135 ture (i.e. the underlying source code). The only differences lie in the exchange grid and the resulting mapping matrices to and

from the model grids. These different mappings are discussed in detail and visualized in the Appendix C. It can be seen from the figure therein that the differences between the intersection-type and ocean-model exchange grid are anticipated to be rather small, see Sect. 3.2, due to the fact that many ocean grid cells are completely contained in a single atmospheric grid cell and therefore undergo the same flux calculations in both settings.

## 2.3 Implementation

As the first step, a working version of the coupled ESM is developed that consists of the MOM5 ocean model and the CCLM atmospheric model. Nevertheless, the ESM is designed such, that other models can be added and the current configuration might be extended or replaced by other suitable models in future. Technically, all components (including the flux calculator) communicate via the widespread OASIS3-MCT (version 4.0) coupling library (Valcke et al., 2013).

Note that since MOM5 and CCLM use time-independent horizontal grids, the exchange grid and all corresponding mapping matrices can be determined once in advance to the model run. Furthermore, the exchange grid is only defined within the coupled region, i.e. the Baltic Sea.

### 2.3.1 Coupling cycle

In the current implementation, the oceanic and atmospheric model exchange the following quantities via the flux calculator during one coupling time step, see also Fig. 5 and for more information on the exchanged variables, see Tab. 1. Note that all exchanged fields are taken instantaneously from the model components and sent to the flux calculator. Currently, no time averaging over a coupling cycle is employed; however, in future work the impact of such an averaging might be considered.

In the beginning of each time step all components have to pass a global barrier, implemented with the Message Passing Interface (MPI) library (Message Passing Interface Forum, 2021), depicted by black vertical bars on the left of the figure. After all components passed the barrier and are thus synchronized, the ocean model starts with updating the internal ice model with the current state information of the underlying water body, while the flux calculator remains in the blocking receive function of the coupling library. In parallel, the atmospheric model does the necessary initialization of the time step until it calls the blocking receive function as well. As soon as the ice model is updated, the ocean model sends its state variables, i.e. surface temperature $T_{s,\nu}$, the albedo $\alpha_\nu$ and the fraction/concentration $f_\nu$ of each surface type $\nu$ (water and 5 different ice classes categorized by their thickness; depicted by double arrows in Fig. 5) from each grid cell to the flux calculator. After the send routine returns, the ocean model is waiting for input to update its ice model from the top. In the meanwhile the sent fields are mapped from the ocean model's grid to the exchange grid and then passed to the flux calculator. With this input, the flux calculator can compute the black-body (thermal) radiation that is emitted by the ocean (see Sect. 2.3.2). This quantity is then sent to both models (and mapped to their grids) where it is added to the atmospheric thermal radiation budget and subtracted from the ocean's one. Note that the thermal radiation that is emitted by the atmosphere is entirely computed in the atmospheric model as it is not simply given as black-body radiation (but also depends on cloudiness and the water vapor in layers above the surface). Since the atmospheric model also requires the ocean's state variables mentioned above (for computing transfer coefficients, radiation fluxes and precipitation), they are passed through the flux calculator to the atmospheric model. However,

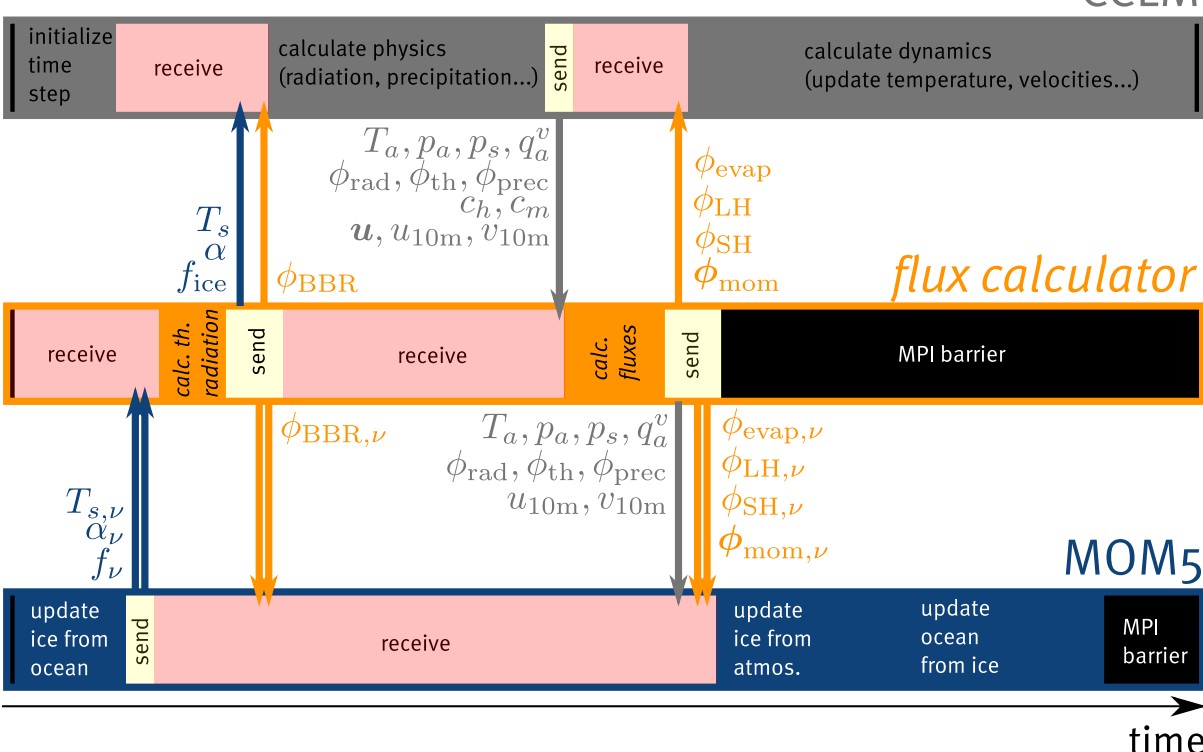

**Figure 5.** Schematic sequence diagram for one coupling time step. The two model components, MOM5 and CCLM, are visualized as the blue and grey blocks, respectively. The flux calculator is represented as the orange block. The calls of the coupling library routines are marked in light red for the blocking receiving function and in light yellow for the non-blocking send routine. Areas with the color corresponding to the particular model (blue and grey) mark the normal operation of the models. The simulation time is runs from left to right. The scaling of the time axis is just illustrative and not quantitatively true. Arrows illustrate the data exchange of the spelled out quantities explained in Tab. 1, where double arrows stand for surface-type dependent fields, i.e. one communicated field for each surface type. The colors of the symbols represent the component from where they originate. For more description see the main text.

due to the fact that the atmospheric model does not distinguish surface categories, these variables are averaged over different
surface types, e.g. $T_s = \sum_\nu f_\nu T_{s,\nu}$, see Tab. 1. By sending these averaged variables from CCLM to the flux calculator, it
is implicitly assumed that above surface of variable temperature, the atmosphere is homogeneous on the scale of the model
grid box, from the first level of the model to the top. This assumption may not be valid in grid boxes with large temperature
gradients, such as those that are partially covered with sea ice. However, it has been a common approach in the state-of-the-art
Earth system models that the atmospheric model component does not treat the surface types separately. Different fluxes over
open ocean and sea ice may be accounted for if the effect of these different surface types is considered separately throughout the
atmospheric column, such as it is the case in the new ICOsahedral Nonhydrostatic (ICON) weather and climate model (Zängl
et al., 2015).

**Table 1.** List of exchanged variables and calculated fluxes.

| Variable | Source | Meaning | Notes |
|---|---|---|---|
| $T_{s,\nu}$ | ocean | surface temperature for surface type $\nu$ | |
| $f_\nu$ | ocean | fraction of surface type $\nu$ | $\sum_\nu f_\nu = 1$ |
| $\alpha_\nu$ | ocean | shortwave albedo for surface type $\nu$ | $0 \leq \alpha_\nu \leq 1$ |
| $T_s$ | ocean | averaged surface temperature | $T_s = \sum_\nu f_\nu T_{s,\nu}$ |
| $\alpha$ | ocean | averaged shortwave surface albedo | $\alpha = \sum_\nu f_\nu \alpha_\nu$ |
| $f_{\mathrm{ice}}$ | ocean | fraction/concentration of ice | $f_{\mathrm{ice}} = \sum_{\nu \neq \mathrm{water}} f_\nu \leq 1$ |
| $\phi_{\mathrm{BBR},\nu}$ | flux calculator | thermal radiation of surface type $\nu$ | treated as radiation of a black body |
| $\phi_{\mathrm{BBR}}$ | flux calculator | averaged thermal radiation | $\phi_{\mathrm{BBR}} = \sum_\nu f_\nu \phi_{\mathrm{BBR},\nu}$ |
| $T_a$ | atmosphere | air temperature at the lowest atmospheric grid cell | |
| $p_a$ | atmosphere | air pressure at the lowest atmospheric grid cell | |
| $p_s$ | atmosphere | air pressure extrapolated to the sea/ice surface | used for calculating the potential temperature |
| $q_a^v$ | atmosphere | water vapor content at the lowest atmospheric grid cell | |
| $\phi_{\mathrm{rad}}$ | atmosphere | downward shortwave radiation flux | separated in direct and diffusive components |
| $\phi_{\mathrm{th}}$ | atmosphere | downward longwave radiation flux | |
| $\phi_{\mathrm{prec}}$ | atmosphere | precipitation flux | separated in rain and snow components |
| $c_h$ | atmosphere | turbulent exchange coefficient for heat/moisture transport | |
| $c_m$ | atmosphere | turbulent exchange coefficient for momentum transport | |
| $\boldsymbol{u}$ | atmosphere | horizontal wind speed vector at the lowest atm. grid cell | separated in east and northward components |
| $u_{10\mathrm{m}}$ | atmosphere | eastward wind speed at 10m height | needed by the internal wave model |
| $v_{10\mathrm{m}}$ | atmosphere | northward wind speed at 10m height | needed by the internal wave model |
| $\phi_{\mathrm{evap},\nu}$ | flux calculator | evaporation flux for surface type $\nu$ | |
| $\phi_{\mathrm{evap}}$ | flux calculator | averaged evaporation flux | $\phi_{\mathrm{evap}} = \sum_\nu f_\nu \phi_{\mathrm{evap},\nu}$ |
| $\phi_{\mathrm{LH},\nu}$ | flux calculator | latent heat flux for surface type $\nu$ | |
| $\phi_{\mathrm{LH}}$ | flux calculator | averaged latent heat flux | $\phi_{\mathrm{LH}} = \sum_\nu f_\nu \phi_{\mathrm{LH},\nu}$ |
| $\phi_{\mathrm{SH},\nu}$ | flux calculator | sensible heat flux for surface type $\nu$ | |
| $\phi_{\mathrm{SH}}$ | flux calculator | averaged sensible heat flux | $\phi_{\mathrm{SH}} = \sum_\nu f_\nu \phi_{\mathrm{SH},\nu}$ |
| $\boldsymbol{\phi}_{\mathrm{mom},\nu}$ | flux calculator | momentum flux vector for surface type $\nu$ | separated in east and northward components |
| $\boldsymbol{\phi}_{\mathrm{mom}}$ | flux calculator | averaged momentum flux vector | $\boldsymbol{\phi}_{\mathrm{mom}} = \sum_\nu f_\nu \boldsymbol{\phi}_{\mathrm{mom},\nu}$ |

After the send function has finished, the flux calculator is waiting for more input from the atmospheric model to calculate the other fluxes. During the blocking of the flux calculator and the ocean model, the atmospheric model can update its physics, i.e. for instance calculating radiation fluxes, precipitation and transfer coefficients. The resulting fields are then sent to the flux calculator, which is released from the blocking receive function, while the atmospheric model is now waiting for the lower

boundary surface fluxes. With the transfer coefficients over the coupled domain (i.e. the Baltic Sea) and the state variables from both model components, the flux calculator can calculate the evaporation, latent and sensible heat as well as momentum fluxes (see Sect. 2.3.2). All quantities are computed on the exchange grid and sent to both models, which can then perform their final updating of the remaining variables with the given surface boundary fluxes. As soon as a component has reached the end of the current time step it is blocked by the MPI barrier before beginning the next cycle. Importantly, calculations in the flux calculator as well as the communication are restricted to grid cells that are coupled, i.e. grid cells that intersect with the ocean model's horizontal grid at the Baltic Sea surface.

In case of the intersection-type exchange grid, the conservation of mass, energy and momentum is naturally ensured in the coupled system. Radiation and precipitation fluxes, that are not computed by the flux calculator, are simply passed through to the ocean model. The downward radiation fluxes are then redistributed by the MOM5 model to different surface types resulting in different net fluxes depending on the particular surface albedo (Sect. 2.3.2). Additionally, the ocean model requires a few atmospheric state variables, i.e. atmospheric pressure and ten-meter wind-speed components for the sea-ice, the turbulence and the wave model that are implemented in the MOM5 component.

### 2.3.2 Flux formulas

The formulas used to calculate the exchanged fluxes are based on the corresponding CCLM (Doms et al., 2011) implementation. This implementation is in turn derived from the work of (Louis, 1979) which is briefly summarized in the following.

Central ingredients of the flux calculation are the air's density $\rho_\nu(x,y,t)$ over the specific surface type $\nu$ (see also Appendix D in the Appendix) and the horizontal wind velocity $\boldsymbol{u}(x,y,t)$ from the lowest atmospheric grid cells. The coefficients $c_h(x,y,t)$ for turbulent moisture and heat transfer as well as $c_m(x,y,t)$ for the turbulent momentum transfer are obtained from the CCLM model via Monin–Obukhov similarity theory (Monin and Obukhov, 1954). Since these coefficients are calculated on the coarser atmospheric grid, they only account for the average sea-ice fraction of the underlying ocean grid cells instead of being decomposed into coefficients over ice and water, which can be very different. Hence, $c_h(x,y,t)$ and $c_m(x,y,t)$ are insensitive to the different scales of the surface heterogeneity, i.e. an ice front separating an ice-covered area from an area of open water (large-scale heterogeneity) is treated in the same way as ice fractured by leads (small-scale heterogeneity). Nevertheless, the presented formulas can be considered as a step forward to improve on this deficiency, since they take different surface categories (water/ice) into account for variables, others than the transfer coefficients, that enter the flux calculation.

The evaporation mass flux is calculated assuming that the air adjusts its water vapor content $q_a^v(x,y,t)$ to the one present at the sea/ice surface $q_{s,\nu}^v(x,y,t)$, i.e.

$$\phi_{\mathrm{evap},\nu}(x,y,t) = c_h \rho_\nu |\boldsymbol{u}| (q_{s,\nu}^v - q_a^v), \tag{1}$$

where all quantities are meant to be functions of $(x,y,t)$, i.e. the horizontal location on the sea surface and time, however, for the sake of brevity we skip these arguments. How the surface-type dependent water vapor content $q_{s,\nu}^v$ is calculated by the flux calculator is shown in Appendix D2.

The latent heat flux is then directly proportional to the evaporation

$$\phi_{\mathrm{LH},\nu}(x,y,t) = \Delta H_\nu \phi_{\mathrm{evap},\nu}, \tag{2}$$

where $\Delta H_\nu$ is the constant for the latent heat consumed/released by either evaporation, freezing or sublimation, depending on the type of phase transition related to the individual surface types.

The sensible heat flux is determined by the difference between the temperatures of the lowest (discretized) atmospheric layer $T_a(x,y,t)$ and the surface $T_{s,\nu}(x,y,t)$, i.e.

$$\phi_{\mathrm{SH},\nu}(x,y,t) = c_h C_p \rho_\nu |\boldsymbol{u}|(T_{s,\nu} - \theta_a). \tag{3}$$

The appearing $\theta_a(x,y,t) = T_a \cdot (p_s/p_a)^{R_d/C_p}$ (where $R_d$ is the gas constant for air) is the atmospheric potential temperature directly at the surface and $C_p$ is the air's heat capacity at constant pressure.

The momentum fluxes (i.e. the shear stress at the components interface) in $x$ and $y$ direction depend non-linearly on the wind velocity $\boldsymbol{u}(x,y,t)$ at the lowest atmospheric layer and are calculated as

$$\boldsymbol{\phi}_{\mathrm{mom},\nu}(x,y,t) = -c_m \rho_\nu |\boldsymbol{u}|\boldsymbol{u}. \tag{4}$$

It is noteworthy, that in the current setup for the Baltic Sea tides are not considered and thus the horizontal velocity components of the ocean's surface ($\propto 10^{-2}$ to $10^{-1}$m/s) are negligible compared to the atmospheric ones ($\propto 1\ldots 10$m/s) and are thus omitted.

The thermal radiation that is emitted in upward direction by the ocean is described by the radiation of a black body having the ocean's surface temperature, i.e. longwave albedo is neglected. Thus the thermal flux can be calculated via the Stephan-Boltzmann law suitable for black-body radiation

$$\phi_{\mathrm{BBR},\nu}(x,y,t) = \sigma T_{s,\nu}^4, \tag{5}$$

where $\sigma$ is the Stephan-Boltzmann constant. Importantly, since the thermal radiation depends strongly non-linear on the temperature, this flux exemplifies the importance of the local consistency within the coupling. In other words, the averaged flux $\sum_\nu f_\nu \sigma T_{s,\nu}^4$ can differ strongly from the flux calculated with the averaged temperature $\sigma \left(\sum_\nu f_\nu T_{s,\nu}\right)^4$, where the latter would correspond to the flux calculated by the atmospheric model.

As stated above, the downward radiation fluxes $\phi_{\mathrm{rad}}(x,y,t)$ (i.e. shortwave and longwave radiation) do not depend only on surface fields; they are thus entirely calculated by the atmospheric CCLM model and then passed through the flux calculator to the ocean model. The ocean model then can use its information on the different albedos $\alpha_\nu(x,y,t)$ of different surface categories $\nu$ to distribute the net flux $\bar{\phi}_{\mathrm{rad},\nu}^{\mathrm{oce}}(\boldsymbol{r},t)$ (without the reflected part) onto the individual surface constituents as

$$\bar{\phi}_{\mathrm{rad},\nu}^{\mathrm{oce}}(x,y,t) = (1 - \alpha_\nu(x,y,t))\phi_{\mathrm{rad}}(x,y,t). \tag{6}$$

The atmospheric model, on the other hand, receives the averaged albedo via the flux calculator from the ocean/ice model as

$$\alpha(x,y,t) = \sum_\nu f_\nu(x,y,t)\alpha_\nu(x,y,t). \tag{7}$$

With this averaged albedo the atmospheric model calculates its own net radiation flux $\bar{\phi}^{\mathrm{atm}}(x,y,t)$ from the downward flux $\phi(x,y,t)$ as

$$\bar{\phi}^{\mathrm{atm}}_{\mathrm{rad}}(x,y,t) = (1 - \alpha(x,y,t))\phi(x,y,t) \tag{8}$$

Although being calculated differently by the two models, it is shown in Appendix D that the resulting net fluxes are equal (when averaging over the contributing surface types in the ocean model grid cell).

Note that all the presented formulas for fluxes might be changed, e.g. to involve different methods, within the flux calculator source code without further changing the model codes. Thus, the presented approach, using an external component for the flux calculation, greatly facilitates sensitivity experiments with respect to surface boundary fluxes. It should be stressed, however, that in order to further improve on the sensitivity of the transfer coefficients with respect to different surface types, the calculation of $c_h(x,y,t)$ and $c_m(x,y,t)$ should be implemented in the flux calculator itself, which is currently out of scope. Alternatively, the employed atmospheric model should be able to treat different surface categories separately and to send the respective fields to the flux calculator.

## 2.4 Simulation setup

The following test setup has been used to perform benchmark simulations. In advance, ERA5 reanalysis data have been prepared as forcing/boundary data for the CCLM atmospheric model for the time period of 1959-01-01 - 1999-12-31. The forcing/boundary data have been processed using the COSMO pre-processing tool *int2lm* (Schättler and Blahak, 2009), which performs the interpolation from the coarse resolution ERA5 data to the employed resolution of the CCLM. With these data the coupled CCLM is forced over the EURO-CORDEX (Jacob et al., 2014) domain using a resolution of 0.22° by 0.22° and parameters similar to the setup used in (Ho-Hagemann et al., 2017, 2020). The coupled MOM5 simulates the Baltic Sea model with a horizontal resolution of 3×3 nautical miles. At the open boundary to the North Sea, we use climatologies for all prognostic model variables. The sea level elevation is estimated from the wind field by a statistical approach and the river runoff and nutrient loads are derived from HELCOM compilations (Neumann et al., 2021). The marine bio-geochemistry is modeled by the latest version of the internally coupled ERGOM (Neumann et al., 2021).

With this setup three runs are performed. First, the intersection-type exchange grid is used (Sect. 2.1). Second and third, the two model grids serve as the exchange grid, respectively.

The runs are performed for a time span of 41 years, i.e. 1959-01-01 - 1999-12-31, where the first year 1959 is considered as spin-up phase. The model time steps are 600s and 150s for the oceanic and atmospheric model, respectively. In the atmosphere, radiation fluxes are updated every hour; the timestep for the physics is 150s and is internally further subdivided to account for fast modes in the dynamics. The ocean model time step is 600s and the time step of the coupling between the two models is also set to 600s to temporally resolve strong wind gusts (Davis and Newstein, 1968). An investigation on the impact of different coupling time step sizes is planned for future work.

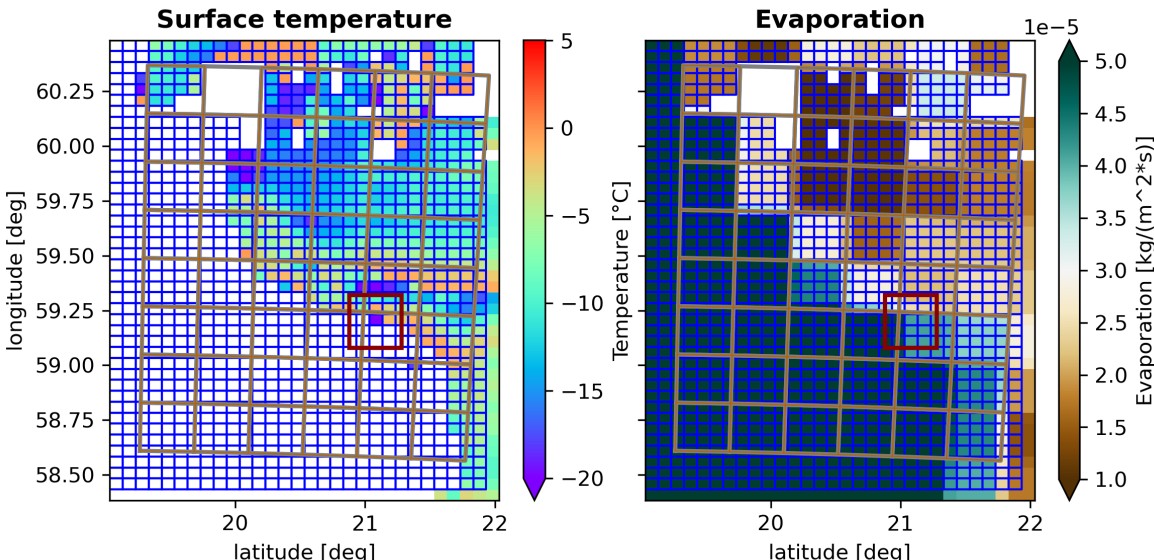

**Figure 6.** Snapshots of ocean's surface temperature (for ice-covered cells) and evaporation flux before the instability, i.e. January 9, 1960 south-east of the Åland island. The left panel shows the surface temperature and the involved grids, i.e. blue boxes represent the ocean's grid and orange boxes the exchange grid that is identical to the atmospheric grid. White ocean cells correspond to ice-free cells which are omitted for the sake of clarity. White areas without boxes represent land. In the right panel the evaporation instead of temperature is depicted. The darkred rectangle is centered around 21.08°E and 59.20°N, see text for further description.

## 3 Results

### 3.1 Instability with atmospheric exchange grid

When using the atmospheric grid as the exchange grid it was impossible to integrate the coupled model over the whole simulation time period. Instead, the model becomes instable after 12 months featuring an unrealistic low surface temperature at specific points on the ocean's grid. Fig. 6 shows a snapshot on January 9, 1960, directly before the model stops. The cause of this instability is exemplified by considering the time evolution of temperature and evaporation for one specific location at 21.08°E and 59.20°N (centered in the darkred rectangle in Fig. 6), where the surface temperature falls below -40 °C within approximately 6 hours (Fig. 7 and magnified in in panel (b) therein). It is evident that the evaporation has increased significantly before the surface temperature starts to vary strongly. Due to the low exchange grid resolution (given by the atmospheric model), the evaporation's magnitude is mainly given by the surrounding ice-free cells (Fig. 6). Thus, the ice-covered cell is cooled down by the loss of latent heat that is determined by the liquid water contained in the ice-free cells. Hence, the instability is a direct consequence of the inconsistency when calculating fluxes on the low-resolution atmospheric grid. Importantly, although the instability is explained with a very specific scenario, such inconsistencies may likely happen if the flux calculation is not consistent with respect to the spatial variation and surface-type dependence of the exchanged quantities.

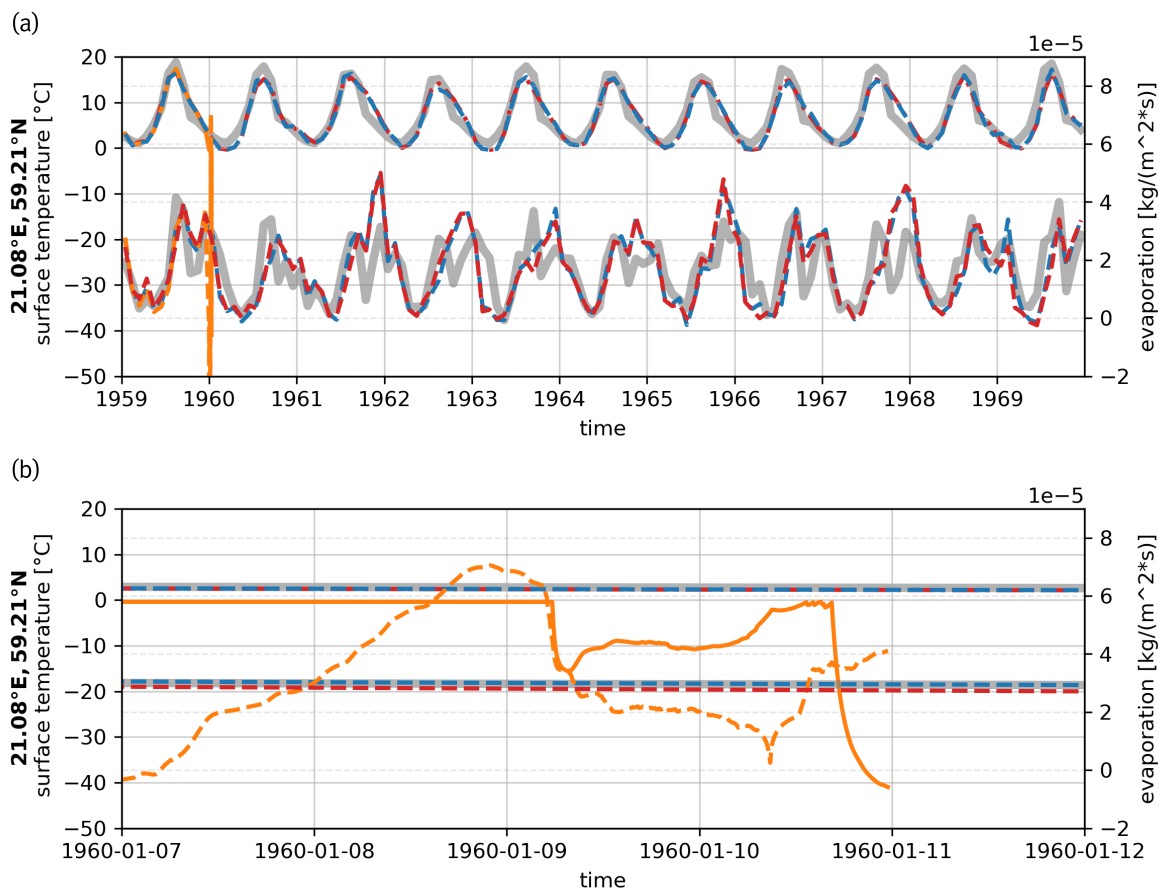

**Figure 7.** Time series of the ocean's surface temperature and evaporation flux at 21.08°E and 59.20°N. The colors represent the different exchange grid types, where blue stands for the intersection grid, red for the ocean model's grid and orange for the atmospheric grid. The light grey curves depict the ERA5 reference data. The upper curves in the upper panel (a) account for the surface temperature (left $y$-axis) whereas the lower curves represent the evaporation (right $y$-axis). The lower panel (b) shows the time evolution when the model using the atmospheric grid gets instable. The solid orange curve shows the surface temperature (left $y$-axis) simulated with the atmospheric exchange grid. The dashed orange line depicts the corresponding evaporation flux (right $y$-axis).

In contrast, the surface temperature and evaporation rates simulated with the ocean-model exchange grid and the intersection-type exchange grid remain stable over the simulation time period. In Fig. 7 one can see that both model types simulate both quantities in good agreement with ERA5 reference data for 21.08°E and 59.2°N. Note that for the sake of clarity only the ten years plus spin-up time (1959) are depicted. A full analysis including a final validation with respect to reliable reference data is out of the scope of this study and will be performed separately in future publications. This would also include the investigation of a higher horizontal resolution of the atmospheric model grid in order to better resolve small call processes, like e.g. cloud

formation, and how theses processes are influenced by the interactive coupling to the ocean variables, e.g. the spatially and time dependent albedo.

## 3.2  Intersection grid vs. ocean model grid

As stated in Sect. 2.2, the differences between using the intersection grid and the ocean-model grid as the exchange grid are anticipated to be small. This becomes also apparent from Fig. 7 since the time series from both model types are almost
identical. Thus the mean difference in the seasonal SST data is not shown directly, as the differences are very small. Instead, in Fig. 8, the Signal-to-Noise Ratio (SNR) of the differences between the SST from both model setups is depicted. The SNR is obtained from the time-averaged seasonal difference $\Delta$SST between the two models divided by their common standard deviation $\bar{\sigma} = \sqrt{\sigma_1^2 + \sigma_2^2}/2$, where $\sigma_{1,2}$ are the individual standard deviations of the seasonal SST from both model runs. It can be seen from Fig. 8, that the mean signals do not differ significantly if one would take e.g. SNR $\geq 2$ as a threshold. The
largest SNR ($> 0.2$) can be observed in the summer season around the Gotland island and in the Gulf of Bothnia, where the appearing pattern will be discussed below.

In contrast, when considering e.g. the differences in the 95-th percentile calculated over the period 1960-01-01 - 1999-12-31, the deviations between the two exchange-grid types are more evident (see Fig. 9). In particular, the SNR for the summer features a similar pattern as for the temporal mean (Fig. 8 second panel from the left) but with much higher values, even above
one. The occurrence of this pattern that is mainly concentrated around the Gotland island might be explained to some extent by the different mappings when using different exchange grids, see Sect. 2.2 and in an extended discussion in the following paragraph. Moreover, for seasons where there is ice (i.e. spring and winter), one can see significant differences (with absolute values $> 1$) in the Gulf of Bothnia. This might be due to surface-type-related inconsistencies as described in Sect. 3.1. We note that many assessments of climate extremes are based on thresholds in the higher percentile temperatures. Thus, the pronounced
difference in the 95-th percentile may indicate that the representation of such extremes will be sensitive to the choice of the exchange-grid. Hence, this may lead to systematic differences in the representation of extremes, such as marine heatwaves which are commonly diagnosed by the through the 90 th percentile SST (Hobday et al., 2016, 2018). If so, one could expect that the intersection-type exchange grid yields more accurate results since it is by construction the most consistent approach. Using a coarser grid could systematically reduce extreme values since the averaging would yield spatially more homogenous
fluxes, in line with what we see in the central Baltic Sea in spring, when the warming (by local shortwave radiation) is fastest. One has to keep in mind though that the simulated atmosphere-ocean systems can show a chaotic behaviour, so marginal differences in the beginning of the simulation may also increase to become random but substantial differences in later points of the simulation. An ensemble simulation with perturbed initial conditions would be needed to tell whether these differences in the representation of extremes are actually systematic.
In order to quantify to the differences between the two exchange-grid types a measure for the mapping consistency is needed. Such a measure can be found in the fraction of a grid cell that is involved for the calculation of the flux that is applied to the grid cell itself. If that fraction is exactly one, then the flux is calculated fully consistently, i.e. no averaging is performed. If the fraction is smaller, then other cells are also impacting the flux calculation, i.e. the information is averaged and fluxes

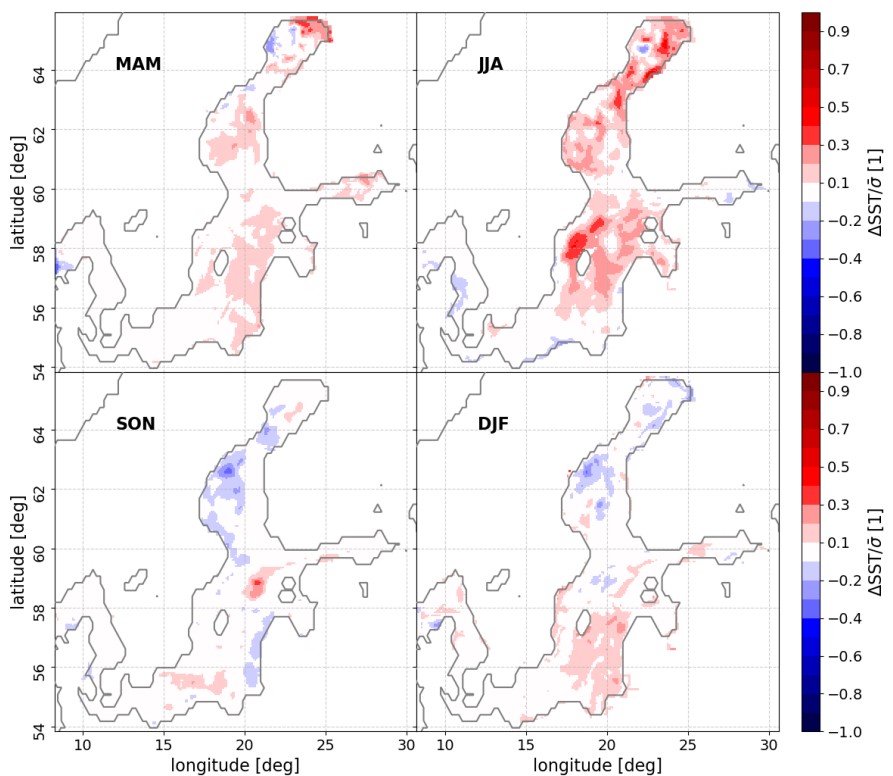

**Figure 8. SNR of differences between the temporal mean SST of ocean-model-grid and intersection-grid simulations.** The maps show the time-averaged difference between the SSTs from the exchange-grid-coupling simulation and ocean-grid-coupling simulation, divided by the standard deviation of the seasonal mean time series. The total evaluation period is 1960-01-01 - 1999-12-31.

may become inconsistent. In Fig. 10, the map of these fractions is depicted for the coupled domain when the ocean's model
grid serves as the exchange grid. Note that the distribution of fractions is additionally convoluted with a Gaussian function
to identify regions, where more inconsistent flux calculation accumulates. This distribution is not isotropic since the involved
model grids have different resolutions and are usually shifted with respect to each other. The resulting pattern can be considered
as the superposition of two stationary waves with different wavelengths and phases yielding a *beating* between the two model
grids (as it can be seen in the lower left panel in Fig. 10).
In the case that the ocean model's grid is used as the exchange grid one has the following situation. First the state information
from both models (corresponding to the two rows in Fig. 10) is mapped to the exchange grid for calculating the fluxes (left
panels therein). If the exchange grid is not the same as the model grid and is not the intersection grid, this mapping involves a
loss of consistency, since the information has to be averaged over several grid cells of the source grid. In other words, the lower

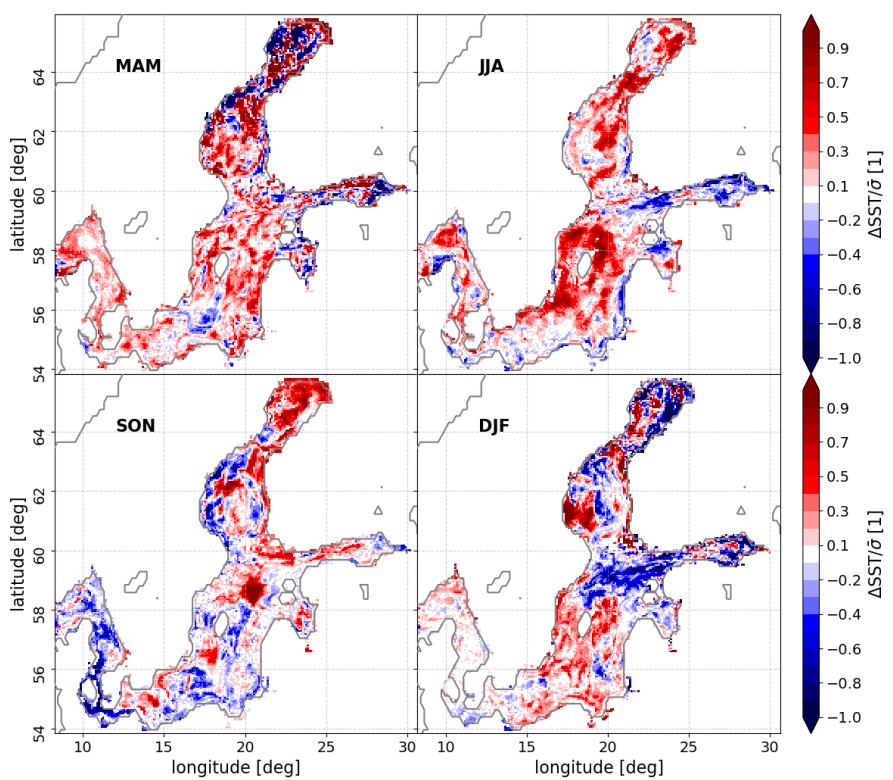

**Figure 9.** SNR of differences between the ocean model's 95-th percentile SST from ocean-model and intersection-grid simulations. See Fig. 8 for further description.

the number of these averaged-over cells is, the higher is the consistency. One can see that the mapping from the ocean to the exchange grid is (by construction) perfectly consistent in the whole coupling domain (upper left panel in Fig. 10). In contrast, when mapping from the atmospheric grid to the exchange grid, there are regions where more atmospheric grid cells contribute to the flux calculation, i.e. grid cells that are averaged over are more abundant. A significant part of these regions is located around the Gotland island and might be related to the pattern one can see in the significance of SST differences in Figs. 8 and 9. Importantly, when using the intersection grid as the exchange grid there is no inconsistency when mapping to it.

In the subsequent part of a coupling step (right panels in Fig. 10) the fluxes are mapped back to the model grids. Again the mapping from the exchange grid to the ocean grid is (by construction) perfectly consistent, whereas the mapping to the atmospheric model grid cells features high inconsistency, since the fluxes are averaged over a large number of exchange-(ocean-)grid cells. Nevertheless, this inconsistency when mapping from the exchange grid to the model grid cannot be avoided with all considered exchange grid types, see Appendix C in the Appendix.

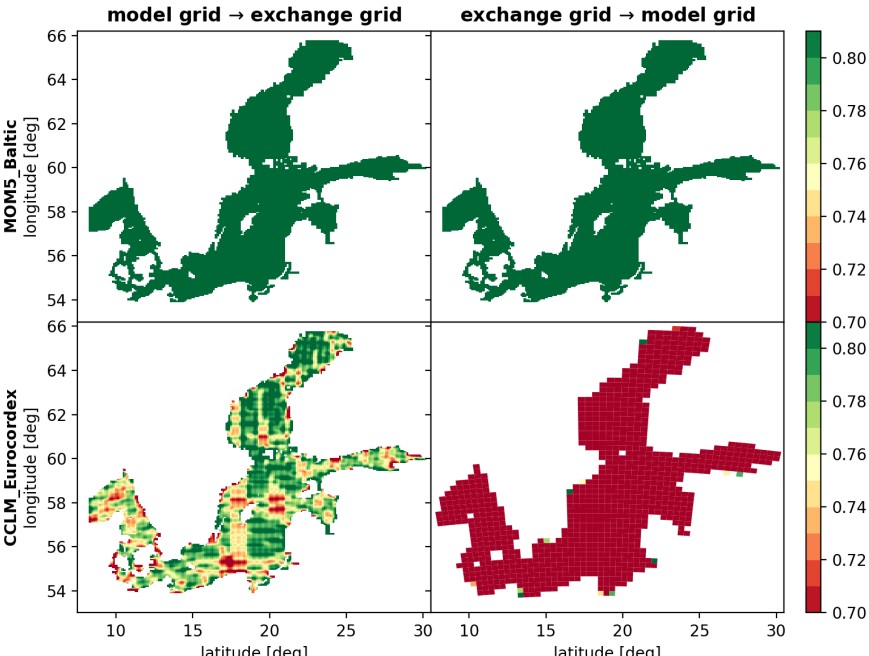

**Figure 10. Consistency map for flux calculation.** Smoothed distribution of grid cell fractions that contribute to the flux calculation. A value of one means that only the grid cell itself contributes to the calculation of fluxes applied to that grid cell. A value smaller than one means that various surrounding cells contribute to the flux calculation.

## 4   Conclusions

The central focus of this article is to present a new regional ESM employing an exchange grid approach and to discuss advantages and disadvantages. The model will be applied in dynamical downscaling experiments.

In contrast to existing coupled RESMs for the Baltic Sea, the presented coupling approach, introduces an extra component that complements the involved circulation model components. This component called the flux calculator computes the fluxes between the models on an exchange grid, formed by the intersections of the model grids. That way, quantities can be exchanged locally consistent and their conservation within in the coupled system is ensured. On top of the aforementioned intersection grid, other possible choices for the exchange grid are presented, i.e. taking one of the model grids as the exchange grid. Importantly, with the developed framework, these different choices can be investigated on the same footing without changing the setup of the involved models.

The current implementation of the coupled ESM consists of the MOM5 model for the Baltic Sea and the CCLM model for simulating the atmospheric dynamics over the EURO-CORDEX domain. The coupling cycle features the mapping of state variables from the models to the exchange grid and subsequent calculation of fluxes via well known formulas introduced in (Louis, 1979) employing Monin–Obukhov similarity theory (Monin and Obukhov, 1954) for obtaining the transfer coefficients.

For each of the aforementioned exchange grid types an individual run of the coupled ESM has been performed with a realistic setup for both model components. In the case the atmospheric model provides the exchange grid, this leads to inconsistencies along borders that separate different surface types, e.g. water to ice or water to land. It turns out that this model configuration is not suitable for the considered setup without further modifications. In fact, the model becomes instable in the second year of simulation as a result of an inconsistent evaporation flux applied to an ice-covered ocean grid cell located at the border to ice-free cells. This is due to the fact that the evaporation is calculated on the coarse atmospheric model grid and thus accounts mainly for the neighboring ice-free cells. It is noteworthy that this deficit might be circumvented by updating the atmospheric model to CCLM version 6.0 or ultimately to the new ICON model (Zängl et al., 2015), which both may account for different surface types. The investigation of an updated configuration of IOW ESM is reserved for future publications.

In contrast, the other two investigated exchange-grid types, i.e. the intersection grid and the ocean model grid case, the model remains stable for the whole integration period 1959-01-01 - 1999-12-31. Both model variants yield seasonal mean SSTs that do not differ significantly. However, the 95-th percentiles of the seasonal SST differ more strongly. The spatial distribution of these differences is related to a consistency map, that reveals regions of inconsistent mapping between the involved grids, when using oceanic exchange grid. In turn, using the intersection grid as the exchange grid naturally avoids these inconsistencies. Whether extreme events are differently described by the various coupling strategies will be investigated in future.

The presented methodology, employing the intersection grid as the natural choice for the exchange grid, provides a consistent treatment of the coupling in climate simulations. For the setup considered here, this approach outperforms the more standard method of calculating fluxes by the atmospheric model on its coarser grid. However, there are no significant improvements apparent compared to the mean sea surface temperatures obtained with a flux calculation that is performed on the oceanic model grid. Whether the more significant differences in the 95-th percentile SST would lead to an improved simulation of extremes remains to be a topic of the next study. Still, the developed framework facilitates a flexible coupling between various components of the Earth system realized by different models. This opens the doorway to deduce robust dynamical downscaling experiments from global climate models to the climate of the Baltic Sea region.

*Code and data availability.* The source code of the IOW ESM (version 1.04.00) is available in multiple repositories collected in the Github organization https://github.com/iow-esm (last access: 27 July 2023). Frozen versions of the code repositories as used for this paper are archived on Zenodo. The main repository (Karsten and Radtke, 2023a) is the entry point for the developed software framework and relates the repositories with each other. The sub repositories are available as Zenodo archives and are listed in the description of the main product. Note that for the CCLM, there is only a patch available that contains the modifications implemented for the IOW ESM. In order to obtain the full model code, the original version of the CCLM code has to be downloaded, further information can be found in https://github.com/iow-esm/components.cclm#readme (last access: 27 July 2023). The same holds for the preparation tool *int2lm*. Note that a complete version 1.01.00 of the coupled CCLM source code and version 1.00.02 of the *int2lm* tool has been made available to the editor and reviewers during the reviewing process of this manuscript.

A frozen version of the IOW ESM manual in *jupyterbook* format is archived on Zenodo (Karsten and Radtke, 2023b). The current online version of the manual can be found at https://sven-karsten.github.io/iow_esm/intro.html (last access: 27 July 2023).

A minimal setup to run the coupled model for a short period of time is stored on Zenodo (Karsten, 2023).

## Appendix A: Acronyms

**CCLM** COSMO model in CLimate Mode

**CESM** Community Earth System Model

**ESMF/NUOPC** Earth System Modelling Framework with the National Unified Operational Prediction

**ESM** Earth System Model

**ICON** ICOsahedral Nonhydrostatic

**IOW** Leibniz Institute for Baltic Sea Research Warnemünde

**MOM5** Modular Ocean Model 5

**MPI** Message Passing Interface

**NCAR** National Center of Atmospheric Research

**RESM** Regional Earth System Model

**SNR** Signal-to-Noise Ratio

**SST** Sea Surface Temperature

## Appendix B: Conservative mapping in the standard coupling approach

In contrast to Fig. 4 in the main text, Fig. B1 depicts how averaged quantities are used to calculate fluxes in the standard approach of coupling, i.e. fluxes are calculated by the atmospheric model.

## Appendix C: Comparing different exchange grids

As stated in the main text in Sect. 2.2 the developed model framework enables comparing different exchange grids on equal footing. The difference between the various setups is discussed in detail in the following in visualized in Figs. C1, C2 and C3. The atmospheric grid is depicted by the grey lines, the ocean model's grid corresponds to the dark blue lines and the exchange grid is shown with the thin orange lines. Exchange grid cells are additionally filled with transparent orange color. The opaque background colors refer to the mean mapping weight contributing to a particular cell on the destination grid. The white numbers show how many cells contribute to that mean. If there is no number given, then only one cell contributes to the particular cell. The columns of each figure correspond to different phases during one coupling time step (from left to right). Left panels depict the sending of state variables from the models to the exchange grid, right panels the communication of fluxes back to the models. The rows account for the two involved models.

The different mapping matrices for different exchange grids can be distinguished at which phase of the coupling spatial averaging is performed. For instance in case of the intersection grid, Fig. C1, the weights are all equal to one when the model's

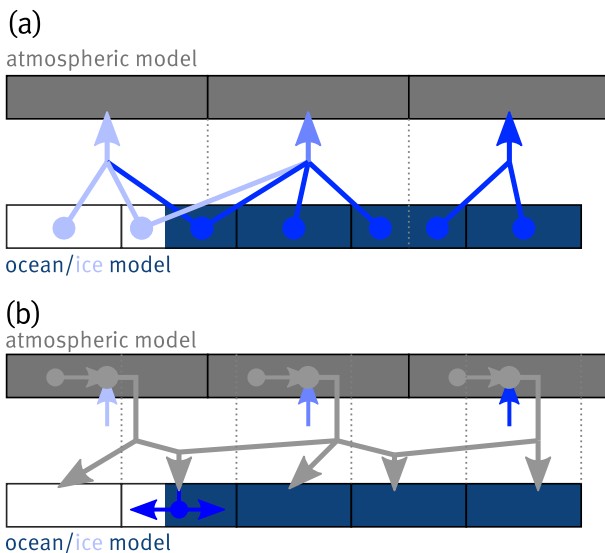

**Figure B1.** The standard way of coupling. Panel (a): Average of ocean's state variables communicated to the atmosphere. Panel (b): Calculation of fluxes in the atmospheric model and remapping on to the ocean model grid.

state variables are mapped to the exchange grid, see white grid cell areas in the left panels therein and note the color bar and figure caption. After the fluxes are calculated from the state variables, the fluxes have to be communicated back to the model grids. This mapping naturally involves averaging over several cells and cannot be avoided since the models do eventually feature different grids. This is illustrated by first the background color of the cells accounting for the mean mapping weight and second by the white numbers that count how many cells contribute to the particular destination grid cell. The higher this counter (and, thus, the lower the average mapping weight is) the more information is lost during the conservative mapping from one grid to the other. Importantly, when using the intersection grid as the exchange grid, the state variables are communicated to the flux calculator without any loss of information. Thus, there is no local inconsistency due to any non-linearity of the flux formulas and no errors stemming from the mapping procedure can be amplified by strongly non-linear dependencies of the fluxes. Averaging over several cells only happens when the fluxes are finally communicated to the models.

In contrast, for the other two cases, there is a loss of information, when the state variables are communicated to the exchange grid for the flux calculation, see Figs. C2 and C3. The case when the ocean model's grid is used, see Fig. C2, seems to be quite similar to the intersection grid case. However, some local information is lost when mapping the atmospheric state variables to the exchange grid before the flux calculation. One can suppose from Fig. C3, that largest local inconsistencies will occur when the standard approach is employed, i.e. the ocean's state variables are first communicated to the atmospheric grid, fluxes are calculated by the atmospheric model and finally the fluxes are communicated back to the ocean. The impact of these inconsistencies is more quantitatively discussed in Sects. 3.1 and 3.2 in the main text.

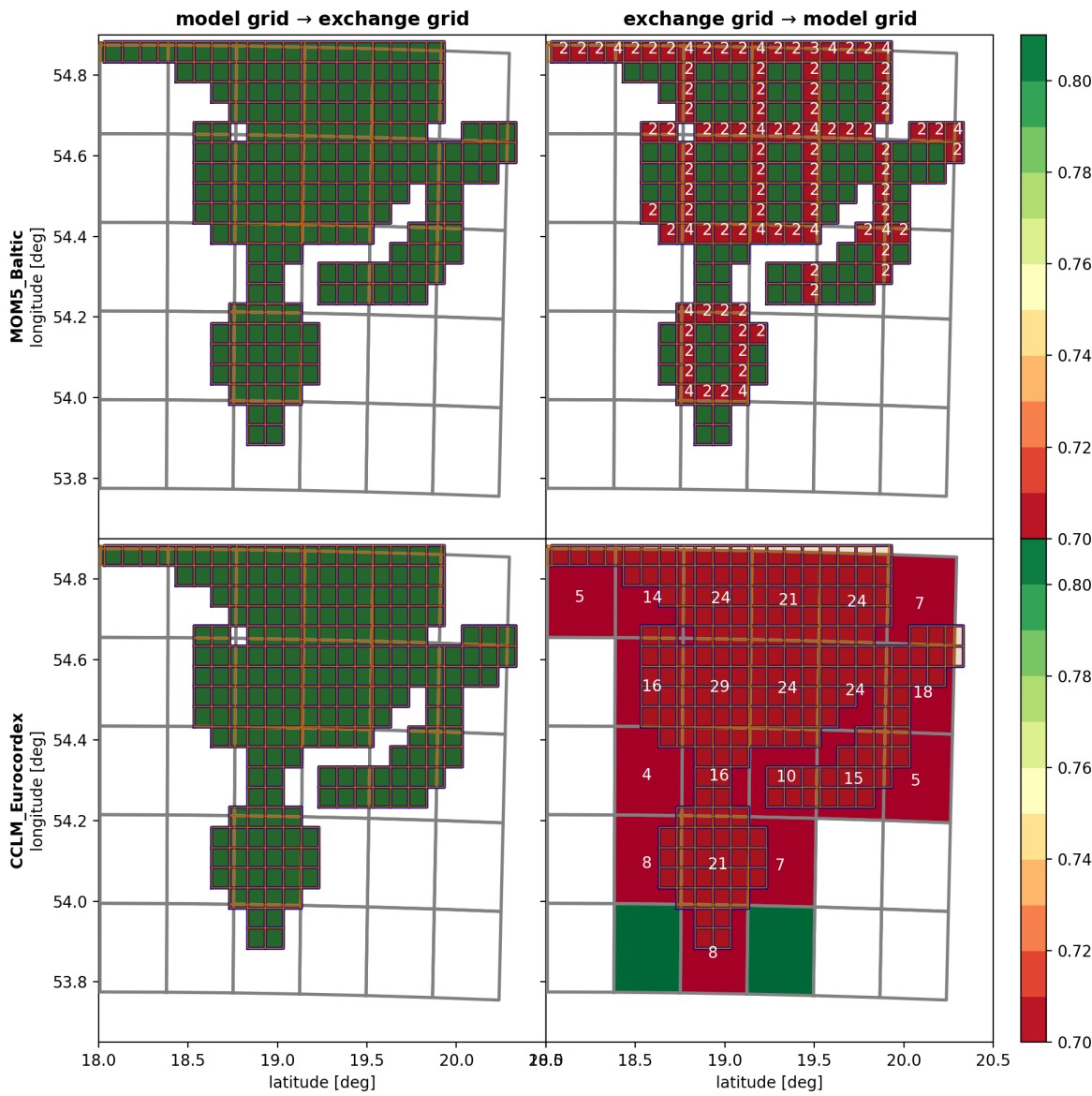

**Figure C1.** Intersection grid is used as the exchange grid. For further description see text.

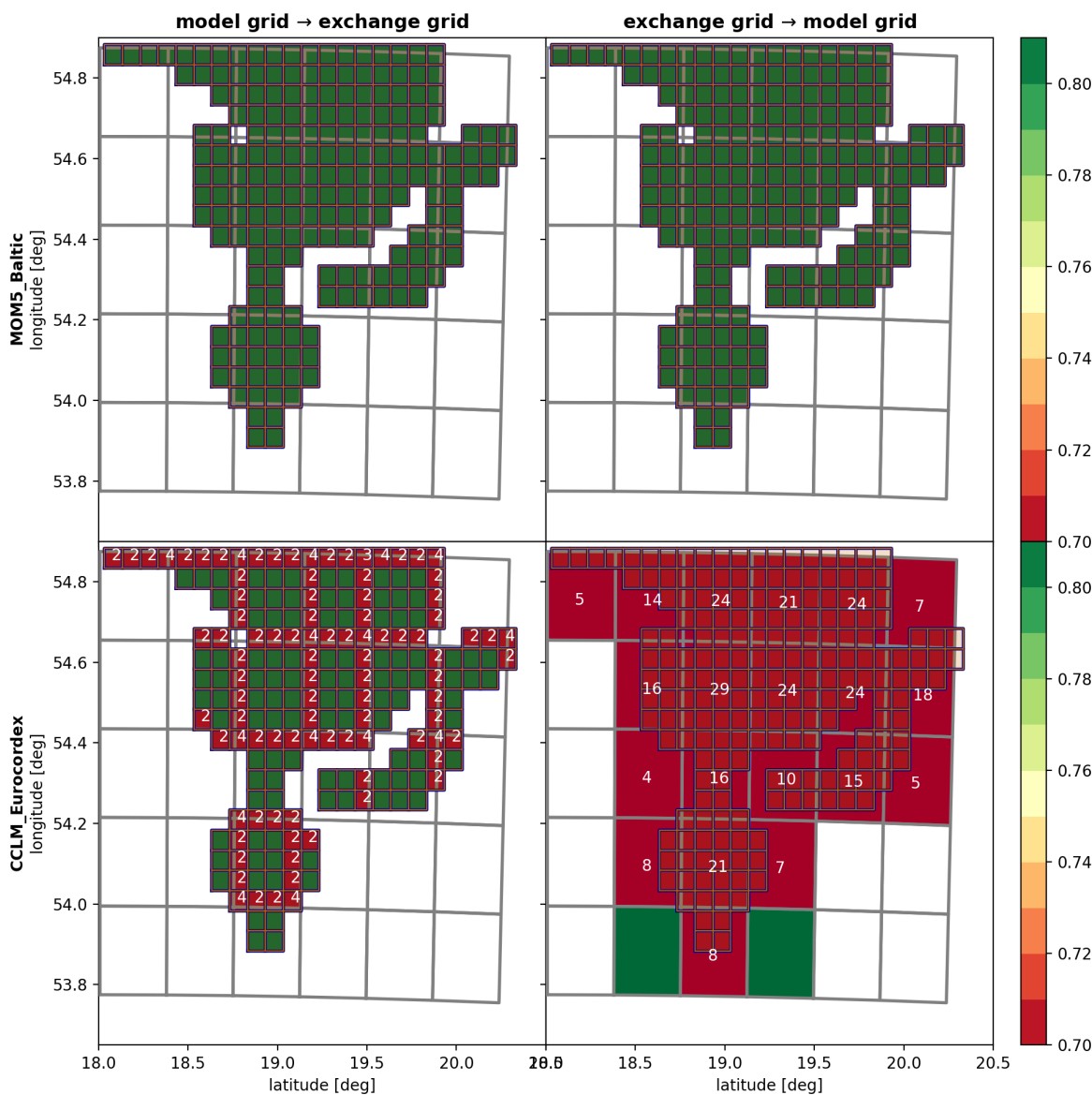

**Figure C2.** Ocean model grid is used as the exchange grid. For further description see text.

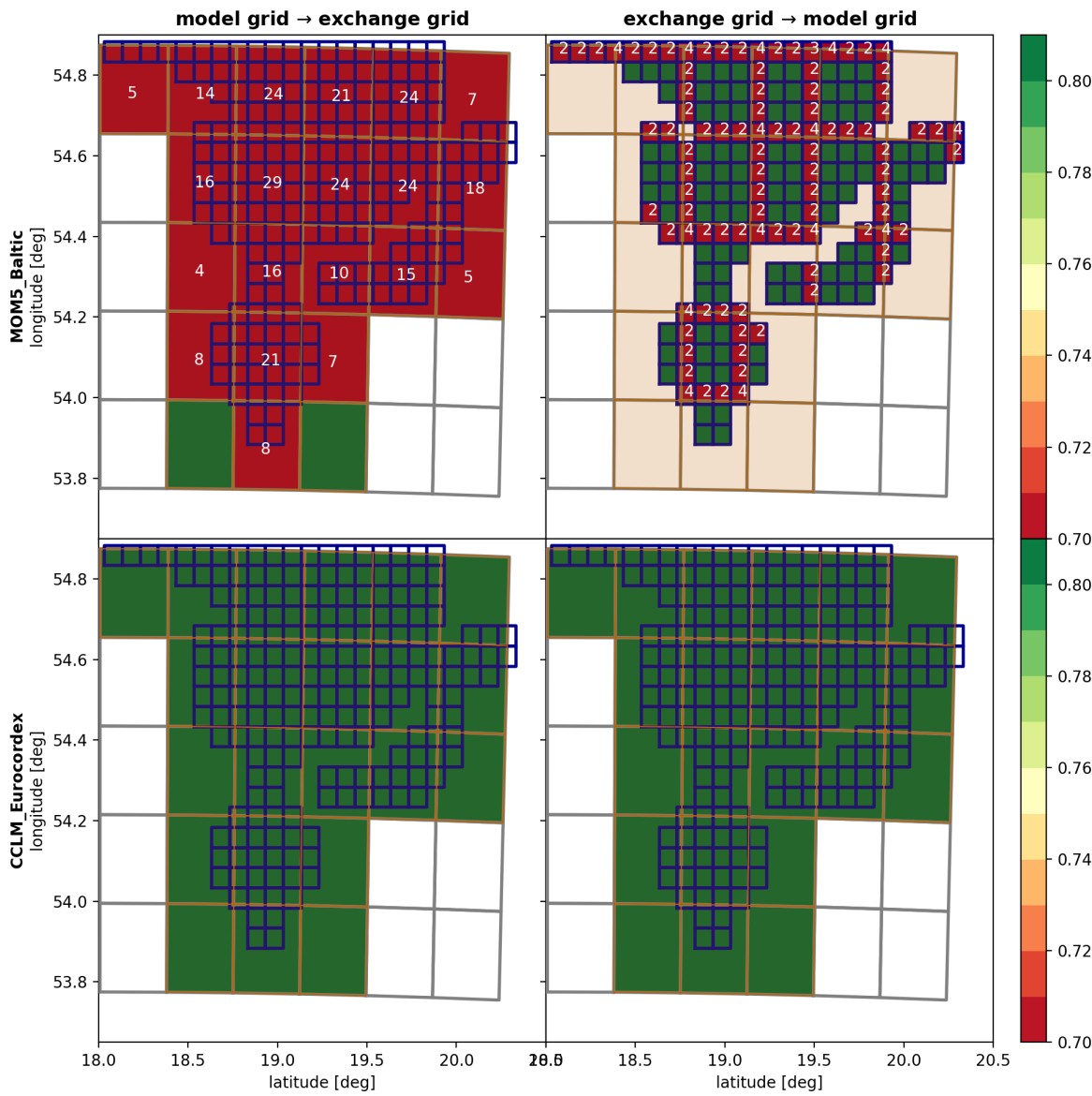

**Figure C3.** Atmospheric grid is used as the exchange grid. For further description see text.

## Appendix D: Flux formulas

### D1  Ingredients of the flux calculation

Using the air pressure $p_a(x,y,t)$ in the lowest atmospheric grid cells then the specific water vapor content $q^v_{s,\nu}(x,y,t)$ over surface type $\nu$ can be calculated via

$$q^v_{s,\nu}(x,y,t) = \frac{R_d/R_v p_{\text{sat},\nu}(x,y,t)}{p_a(x,y,t) - (1 - R_d/R_v)p_{\text{sat},\nu}(x,y,t)}, \tag{D1}$$

with the gas constants $R_d$ for dry air and $R_v$ for water vapor. The sea-surface temperature $T_{s,\nu}(x,y,t)$ determines the saturation pressure $p_{\text{sat},\nu}(x,y,t)$ that is calculated according to the Tetens approximation (Tetens, 1930) and the extension to temperatures below zero by Murray (1967), i.e.

$$p_{\text{sat}}(x,y,t) = 0.61078 \cdot \exp\left(\frac{\beta_\nu \cdot T_{s,\nu}(x,y,t)}{T_{s,\nu}(x,y,t) + T_\nu}\right) \tag{D2}$$

with $T_{s,\nu}$ in °C and where $\beta_\nu = 17.27$ if $\nu$ corresponds to water and $\beta_\nu = 21.87$ for ice. The temperature parameter $T_\nu = 237.30$°C for water and $T_\nu = 265.50$°C for ice.

Having the water vapor content $q^s_{v,\nu}(x,y,t)$ at hand, one may then calculate the temperature $\tilde{T}_\nu$ at which dry air at the surface would show the same energy $p \cdot V$ as the moist air which is there now.

$$\tilde{T}_\nu(x,y,t) = T_{s,\nu}(x,y,t)\left(1 + (R_v/R_d - 1)q^v_{s,\nu}(x,y,t)\right) \tag{D3}$$

This temperature is related to the air's density by the ideal gas law (valid for dry air)

$$\rho_\nu(x,y,t) = \frac{p_a(x,y,t)}{R_d \tilde{T}_\nu(x,y,t)} \tag{D4}$$

With the density $\rho_\nu(x,y,t)$ all considered surfaces fluxes can be calculated as presented in the main text, Sect. 2.3.2.

### D2  Calculating net radiation fluxes for different surface types

As it was stated in the main text, the net radiation fluxes are calculated differently in the atmospheric and oceanic models. However, if we compare both net fluxes by taking the mean over the different surface types considered by the ocean model, we observe

$$\sum_\nu f_\nu(x,y,t)\bar{\phi}^{\text{oce}}_{\text{rad},\nu}(x,y,t) = \tag{D5}$$

$$\sum_\nu f_\nu(x,y,t)(1 - \alpha_\nu(x,y,t))\phi_{\text{rad}}(x,y,t) =$$

$$(1 - \alpha(x,y,t))\phi_{\text{rad}}(x,y,t) \equiv \bar{\phi}^{\text{atm}}_{\text{rad}}(x,y,t).$$

Thus, the net fluxes calculated by both models are equal if we consider the mean flux applied to an ocean grid cell covered by different surface categories.

*Author contributions.* The code, scripts and manual of the IOW ESM in its current state have been developed by SK. HR designed the exchange grid approach including the flux calculator and implemented first prototypes of the coupling code and running scripts. In addition, HR guided the further development of the software framework. HTMH-H implemented a first version of the coupling interface to the CCLM. The setup for CCLM was provided by HTMH-H and the setup for MOM5 was provided by TN. Atmospheric boundary condition generation was done by HM. HM also contributed to the discussions leading to a qualitative improvement of the model results. The coupled model simulations were performed, analyzed and processed by SK. The manuscript text (including figures) has been prepared by SK and further iterated with the support of MG, HEMM and all other authors. HEMM designed and coordinated the research.

*Competing interests.* The authors declare that they have no conflict of interest.

*Acknowledgements.* The research presented in this study is part of the Baltic Earth programme (Earth System Science for the Baltic Sea region, see http://www.baltic.earth) and was funded by the Federal Ministry of Education and Research (Bundesministerium für Bildung und Forschung, BMBF) through the project CoastalFutures (03F0911E). HTMH-H acknowledges the German REKLIM project. The authors gratefully acknowledge the computing time granted by the Resource Allocation Board and provided on the supercomputer Lise and Emmy at NHR@ZIB and NHR@Göttingen as part of the NHR infrastructure. The calculations for this research were conducted with computing resources under the project `mvk00050`. The authors also thank Christian Steger, manager of the CLM community, for his help in publishing parts of the source code.

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
