# Peer review of "Flux coupling approach on an exchange grid for the IOW Earth System Model (version 1.04.00) of the Baltic Sea region"

_Geoscientific Model Development, 2023_

## Author Comment (AC1)

**Reply on RC1**

**Sven Karsten**

**November 21, 2023**

In the following we address the reviewers concerns point by point by first repeating the comment in italics followed by our reply and a description of changes performed in the revised manuscript in red.

**1 Major concerns**

*"I didn't really succeeded to catch the time pattern of exchanges. Fig. 5 is supposed to address this issue, but it doesn't explain how exchanges are synchronised with models time stepping, and the frontiers of the time steps. We need a time sketch showing when the different components (models and flux calculator) are working, when they are waiting, etc ..."*

We agree to the reviewers concern that the details of the coupling cycle with respect to the time behavior of the individual components are not sufficiently represented in Fig. 5.

We extend the figure, such that is clearly visible where the models perform there original computations and where they are blocked by the communication to or from other components. Further details are then given in the text section *2.3.1 Coupling cycle.*

*"In Fig. 2, Fig. 3 and Fig.4, an ocean grid cell is either ice free or fully ice covered. But line 145 reads that fluxes are computed for free ocean and each ice category. Do you consider partially ice covered ocean grid boxes, with several ice categories ? Is so, please show ice fraction and ice categories (by thickness classes ?) in the figures, and give more details in the text about the flux computation over different surfaces."*

We do consider partially ice-covered ocean grid cells. The employed MOM5 ocean model distinguishes between liquid water and 5 different ice classes related to the thickness of the ice. The fluxes that are sent to the ocean model are individual fields for each ice class which are then differently treated in the model code.

However, we agree that this can be misunderstood from figures 2, 3 and 4, where the ocean grid boxes look either fully ice covered or entirely ice free.

We change figures 2,3 and 4 such that it becomes clear that the ocean grid cells can be partially covered with ice. However, for the sake of clarity, we omit the visualization of different ice classes in these figures but rather make a clear statement in the text that different ice categories are considered by the employed ocean/ice model.

*"Radiation is not computed by the flux calculator. But each ocean grid cell may have different albedos, especially if there is sea ice. Do all ocean grid cells receive the same short wave flux ? Or do you use the albedos of each cell and surface type (water, ice) to redistribute the solar flux ? With no redistribution, the flux can be very unrealistic (for instance very low solar flux toward ocean when the ice fraction is large, very high solar flux on ice when ice fraction is low). Please details that, and address the potential impact of your procedure."*

It is true that the flux calculator does not calculate the radiation fluxes. Instead the *downward* radiation fluxes are passed via the flux calculator from the atmospheric to the ocean model. The ocean model then can use its information on the different albedos of different surface categories to distribute the *net* flux (without the reflected part) onto the individual surface constituents.

The atmospheric model, on the other hand, receives the averaged albedo via the flux calculator from the ocean/ice model. With this averaged albedo the atmospheric model calculates its own net radiation flux from the downward flux. It can be shown that the net fluxes calculated by both models are equal if we consider the mean flux applied to an ocean grid cell covered by different surface categories.

We agree to the reviewer that this consideration is not apparent in the manuscript.

We address this discussion on the net radiation fluxes that are calculated by the individual models in section *2.3.2*

*Flux formulas* of the revised manuscript and will give additional formulas in the supplement. Moreover, in order to provide all the details of the flux calculation we complete the notation of all variables and all flux formulas such that it becomes visible, which of them refer to different surface types. This is done by introducing the index $\nu$ that labels water and the different ice classes.

**2  Minor concerns**

*"Line 59 "by (Balaji et al. 2006)" -> "by Balaji et al. (2006)""*

The citation is corrected accordingly.

*"Line 174, 183, 188. The atmospheric wind speed is used in the bulk formulae. In Fig. 5, exchanged variables include Uair, Vair, U10m, V10m. What is the difference, and what variables are used in which flux computation ?"*

The intention behind the full names in the figure was to simplify the understanding of the figure at a first glance. However, we agree that the connection between these names and symbols in the flux formulas is not exactly clear.

In the revised manuscript we harmonize the names of the variables in the figure with those from the formulas. In order to still provide fast understanding of the figure, we add a table where all appearing symbols are explained. Together with the revision due to major concern 3 this gives a more rigorous presentation of the involved variables and calculations.

*"Line 189 : "It is noteworthy, that the horizontal velocity components of the ocean's water body are negligible compared to the atmospheric ones and are thus omitted". That's true if the model does not resolve the tides. Is it the case ?"*

In the current setup tides are indeed not resolved, since they do not play a significant role for the Baltic Sea. Thus, the velocity components of the ocean's surface are in the order of $10^{-2}$ to $10^{-1}$ m/s and the atmospheric wind speed is typically one to three orders of magnitude larger. Hence the relative velocity of the wind to the water surface can be safely approximated by the wind speed only.

On the other hand, to include the water velocity in the flux formulas would be indeed possible by sending this information from the ocean model to the flux calculator. However, for historical reasons, we started the implementation of the flux calculation with the formulas implemented in the CCLM, where the water velocity components are simply neglected.

We add the comparison between the order of magnitudes of the different velocities to the revised manuscript and note that tides are not taken into account.

*"Line 200 : "Moreover, the presented formulas might be updated to more elaborate schemes using more sophisticated theories e.g. a TKE-based ansatz for the calculation of transfer coefficients (Doms et al., 2011)." As the transfer coefficients are computed in the atmosphere, and not in the flux calculator, it seems not feasible to implement a TKE formulation in the flux calculator. "*

The intention of this statement was to underline which possibilities are facilitated by the presented framework. However, we agree to the reviewer that this particular "add-on" is currently too elaborate for the flux calculator.

We modify the statement in the revised manuscript and give a more general perspective on how to experiment with different formulations of the flux calculation.

*"Line 219 : time steps are 600s and 150s for the oceanic and atmospheric model, respectively. For the atmosphere I suppose that this is the 'physics' time step, and advection is called more frequently ? A question important for the coupling is the time step of the computation of solar radiation. Is it called at the same pace than the coupling ? Or with a longer time step ? "*

With the time step 150s the physics part (e.g. horizontal diffusion, tracer advection etc.) is updated. The dynamical core (advection of the velocity field) is updated more frequently with a smaller time step that is internally calculated from the given 150s to obey the CFL criterion for horizontal acoustic wave propagation via a third order Runga-Kutta

scheme. The radiation is only updated every our. Since our coupling time step is 600s, the radiation is updated only every 6th coupling cycle. The small coupling time step of 600s has been chosen to resolve strong wind gusts as it is written in the manuscript. An investigation of the impact of the coupling time step size is planned for future publications.

We detail the description of the time stepping of the atmospheric model in *Sect. 2.4 Simulation setup* of the revised manuscript and provide an outlook for future work including an investigation of different coupling time steps.

---

## Author Comment (AC2)

**Reply on RC2**

Sven Karsten

November 21, 2023

In the following we address the reviewers concerns point by point by first repeating the comment in italics followed by our reply and a description of changes performed in the revised manuscript in red.

**1  Minor clarifications needed**

*"l 189: I am surprised that ocean velocity is not coupled. Could you give some more justification for this. Adding these should not have a big impact on the run time of the model. Could you provide some numbers to explain why the expected error is negligible."*

Since tides do not play a significant role for the Baltic Sea, the velocity components of the ocean's surface are in the order of $10^{-2}$ to $10^{-1}$ m/s and the atmospheric wind speed is typically one to three orders of magnitude larger. Hence the relative velocity of the wind to the water surface can be safely approximated by the wind speed only.

On the other hand, to include the water velocity in the flux formulas would be indeed possible by sending this information from the ocean model to the flux calculator. However, for historical reasons, we started the implementation of the flux calculation with the formulas implemented in the CCLM, where the water velocity components are simply neglected.

We add the comparison between the order of magnitudes of the different velocities to the revised manuscript and note that tides are not taken into account.

*"l 197-199: It is not clear to me what is happening with the radiation fluxes w.r.t. the exchange grid. I stumbled over the phrase "do not simply depend on". Perhaps you could rephrase this paragraph to make it better understandable."*

To our best knowledge, the calculation of downward radiation fluxes cannot be reasonably implemented via bulk formulas that take only surface fields as input. Thus, the calculation of such fluxes is currently out of scope of the flux calculator's capabilities which uses bulk formulas to calculate mass, momentum and heat fluxes. Consequently the downward radiation fluxes are still entirely calculated by the atmospheric model and then sent via the flux calculator to the ocean model.

We rephrase the corresponding paragraph in Sect. *2.1 The exchange grid and the flux calculator* to meet the wording given above.

*"l 220: Please clarify in the text if atmosphere coupling fields are averaged in time over one coupling interval."*

The coupling fields sent from the oceanic and atmospheric models to the flux calculator are instantaneous fields and hence not averaged over a coupling time step. I might be a future feature to be able to switch between instantaneous and averaged fields and to investigate the impact of both variants.

In the revised manuscript we state in Sect. *2.3.1 Coupling cycle* that the fields are not averaged over time and give a perspective for future investigations.

*"Either as part of the introduction or if you decide for adding a discussion section: Could you indicate how your flux calculator approach differs from the "Flux Coupler" in CESM2.1: https://www.cesm.ucar.edu/models/cpl which is also programmed to calculate fluxes."*

The approach used in the CESM2.1 model is indeed very similar to our flux calculator. However, in contrast to our regional model, the CESM is designed for for global climate modeling as it is also part of the CMIP6 ensemble.

We thank the reviewer for pointing out the similarity between the coupling approaches employed in our model and in the CESM and added a corresponding sentence and citation in the introduction.

**2 Technical corrections**

*"l 63 and elsewhere in the text: suggestion to a use some other word than "automatically" or simply drop it."*

We replace "automatically" by "naturally" in cases our conservation of quantities in our approach is addressed and by "on-the-fly" in the introduction of the revised manuscript.

*"l 199: only → Only"*

The sentence followed by "only" was accidentally left in the manuscript and is thus dropped in the revised manuscript.

**3 Figures**

*"Fig 1: increase size if axis labels*
*Fig 4: increase font size for text in the figure*
*Fig 5: increase font size for text in the figure*
*Fig 8: increase size of figures and font size of axis labels*
*Fig 9: increase size of figures and font size of axis labels"*

We follow the reviewer's request and increase the font sizes and sizes of the mentioned figures. For figures 8 and 9 we thus dropped the total time average which was anyway not discussed in the text.

**4 Some suggestions and thoughts (optional) for a discussion section**

*"Could "Flux Coupling" find its way into the title as both the exchange grid and the flux coupling are somewhat key here?"*

We agree to the reviewer that both, the exchange grid and the flux coupling, are at the heart of the presented approach and thus both deserve to be part of the title.

We suggest to change the title of the revised manuscript as "Flux coupling approach on an exchange grid for the IOW Earth System Model (version 1.04.00) of the Baltic Sea region".

*"Couldn't you finalise the conclusions with some strong point, e.g. what you gained and what has been significantly improved in your model? I find it a bit disappointing being told in the final message only what you would like to do in future."*

We agree to the proposal of the reviewer and added a small paragraph at the end of the revised manuscript.

*"While the improvements you show are convincing I wonder how this relates to a model configuration where the atmosphere land ocean are run on identical grids – at the same resolution as your ocean?"*

We want to draw the attention of the reviewer to the first two paragraphs of the introduction, where we address the feasibility of having the same grid for ocean and atmospheric models. As it is stated there, the Baltic Sea requires a particularly high spatial resolution due to the "complicated coast lines given by numerous islands, narrow channels between the basins and the small baroclinic Rossby radius [...]. However, the corresponding atmospheric circulation is usually simulated on a much larger domain, since the pathways of cyclones originating from the North Atlantic region should be part of it. For this reason, the atmospheric model cannot be discretized with the same high resolution as the ocean model at reasonable numerical costs. ". This was used as the motivation to develop a different strategy as it is carried out in the manuscript.

*"As you couple albedo how would small scale processes, in particular clouds, would change the game if better resolved with a higher horizontal resolution of the atmosphere model. On the coarse atmosphere grid some horizontal averaging is done by construction."*

Investigations of such a kind will be done for the planned follow-up publication including the validation of the model with higher horizontal resolution of the atmospheric grid.

We add a sentence for giving this kind of outlook in Sect. *3.1 Instability with atmospheric exchange grid* in the revised manuscript.

*"How would implicit coupling change the game. See e.g. Kang et al., 2021: Mass-conserving implicit–explicit methods for coupled compressible Navier–Stokes equations. https://doi.org/10.1016/j.cma.2021.113988 or Balaji et al., 2016: Coarse-grained component concurrency in Earth system modeling: parallelizing atmospheric radiative transfer in the GFDL AM3 model using the Flexible Modeling System coupling framework. https://doi.org/10.5194/gmd-9-3605-2016"*

Both mentioned publications deal with the optimization of the coupling also within the employed model components and their respective source code. For instance, in Kang et al., 2021 it is proposed to advance one of the components (the "stiffer one, i.e. the ocean) implicitly and the other (i.e. the atmosphere) explicitly. In Balaji et al., 2016, the concurrent treatment of the radiation and the rest of the atmospheric time step is used to optimize the performance of the coupled model. We currently see these strategies far beyond the scope of our manuscript, which introduces the flux coupling treated by an additional executable, the flux calculator, acting on an exchange grid. One key point of the presented philosophy was to keep the involved model components and their source code as close to the original as possible, in order to permit flexibility in choosing these components. A concrete investigation of how to optimize the overall performs of the coupled model will surely be a topic for future publication, where the work of Kang and Bajali will be certainly referenced.

*"How does your approach relate to the one in the Bergen Model described by Furevik et al, 2003: Description and evaluation of the Bergen climate model: ARPEGE coupled with MICOM. https://doi.org/10.1007/s00382-003-0317-5"*

The given reference by Furevik et al, 2003 presents an interesting strategy to overcome the same issues that are tackled by our approach. The method is based on a Monte Carlo method, where a large number of points are distributed over both (oceanic and atmospheric) model domains. The fraction of how many points that are contained in one atmospheric grid cell *and* also in an underlying ocean model grid cell is then stored and used for the mapping of exchanged quantities. To our understanding, this fraction would ultimately converge to the area fraction of one of our exchange grid cells and the linked atmospheric grid cell, if the number of random points approaches infinity. Thus, we view the method presented in Furevik et al, 2003 as an alternative but equivalent implementation of the exchange grid. However, it does not feature the calculation of the fluxes on the exchange grid. Still, it is correcting fluxes on the higher resolved oceanic grid with the employed subgrid interpolation. In how far this compares to our flux coupling is not so easily deducible.

We will refer to Furevik et al, 2003 in the introduction of the revised manuscript as an additional example of an exchange-grid-like approach.

---

## Editor Decision (ED1)

December 8[th] 2023

Dear author,

Thank you for your revised manuscript, into which you answered many comments of the 2 reviewers. However, I think there are still few issues that were not addressed and I would like you to consider the following remarks.

1.  Missing conclusion regarding the added value of the exchange grid

You have added a paragraph in the conclusion insisting on the flexibility of the exchange grid approach. However, I think that one important conclusion, i.e. that calculating the fluxes on an exchange grid does not improve the results compared to calculating them on the ocean grid, at least when the ocean is much finer than the atmosphere as in your case, should also be stressed.

2.  Main first comments of Referee #1.

I think you did not considered these comments that are :

- « 1/ In this coupling scheme, there is the assumption that even above surfaces of variable temperature, the atmosphere is homogeneous, on the scale of a model grid box, from the first level of the model to the top. This is a very strong assumption, and in fact a priori false. You only have to look at the clouds above a free water-sea ice front to that at least the planetary boundary layer may have very different vertical structure within one atmosphere grid box. »
- « 2/ The transfer coefficients ch and cm are sent by the atmosphere (Fig. 4). They are not calculated by the flux calculator. It is quite possible in reality to have stable air above the ice and unstable air above the open water. This yield very different transfert coefficients above the two surfaces. This should lead to very different flux compare to what the flux calculator computes. Note that if the calculations of these coefficients involve iterations in the atmosphere, this can indeed be complicated to compute them in the flux calculator. »

Referee #1 wrote : « This (sic) two limitations should be explained to the reader, with if possible some descriptions of the potential impact, and technical constraints that explained the chosen algorithm. » and I don't see anywhere where this has been done.

Also the method does not separate scales. It treats an ice front separating an ice-covered area from an area of open water (large-scale heterogeneity) in the same way as ice fractured by leads (small-scale heterogeneity).

Please add something to address these remarks.

3.  Paragraphs lines 112-15 and lines 226-228

The sentence « The calculation of some fluxes cannot be reasonably implemented via bulk formulas that take only surface fields as input » sounds awkward to me, as does the sentence « do not only simply depend on surface fields » line 227 as noticed by Referee #2.

Please consider rephrasing lines 112-113 more clearly with something like "However, the calculation of some fluxes does not depend only on surface fields and therefore are out of scope of the flux calculator capabilities. In particular, …"

On line 114, consider changing "will still be calculated" with "are calculated".

Please consider rephrasing lines 226-228 as: "As stated above, the downward radiation fluxes $\varphi rad(x,y,t)$ (i.e. shortwave and longwave radiation) do not depend only on surface fields; they are thus entirely calculated by the atmospheric CCLM model and then passed through the flux calculator to the ocean model."

4.  Paragraph lines 239-242

The first and last sentences of this paragraph are redundant. Please consider merging them into "Note that all the presented formulas for fluxes might be changed, e.g. to involve different methods, within the flux calculator source code without further changing the model codes. Also, I think you should take the opportunity there to stress the fact that this is not true for the transfer coefficients $ch$ and $cm$, which are calculated and sent by the atmosphere, see my remark 1. above.

5.  Paragraph lines 257-261

With your latest modifications, this paragraph does not read well. Please consider rephrasing for something like: "In the atmosphere, radiation fluxes are updated every hour; the timestep for the physics is 150s and is internally further subdivided to account for fast modes in the dynamics. The ocean model time step is 600s. The time step of the coupling between the two models is also set to 600s to temporally resolve strong wind gusts (Davis and Newstein, 1968). An investigation on the impact of different coupling time step sizes is planned for future work.

6.  Reference to the appendix

For clarity, change the references to the appendices using the word "appendix" and not "Sect." and give the number of the appendix. For example, line 89, "Sect. B" should be "Appendix B" and at line 192 "Sect. D in the Appendix" should be "Appendix D2".

7.  Line 153

Consider changing « … is employed, however, … » for « … is employed; however, … »

---

## Author Response (AR2)

**Reply to the public justification**

**Sven Karsten**

**December 18, 2023**

We thank the reviewers and editor for further suggestions that greatly improve the manuscript. In the following each of the remaining issues is addressed by first repeating the remark in italics, followed by our reply and the changes performed for a revised manuscript in red.

*"1. Missing conclusion regarding the added value of the exchange grid*
*You have added a paragraph in the conclusion insisting on the flexibility of the exchange grid approach. However, I think that one important conclusion, i.e. that calculating the fluxes on an exchange grid does not improve the results compared to calculating them on the ocean grid, at least when the ocean is much finer than the atmosphere as in your case, should also be stressed."*

We added two sentences to the conclusions (lines 389-392) stating that there is no apparent improvement for mean sea surface temperatures and that the impact of the differences in the 95-th percentile is still unknown.

*"2. Main first comments of Referee # 1.*
*I think you did not considered these comments that are :*
*• 1/ « In this coupling scheme, there is the assumption that even above surfaces of variable temperature, the atmosphere is homogeneous, on the scale of a model grid box, from the first level of the model to the top. This is a very strong assumption, and in fact a priori false. You only have to look at the clouds above a free water-sea ice front to that at least the planetary boundary layer may have very different vertical structure within one atmosphere grid box. »*

We apologize that we have overseen these comments in the first revision. To the first comment, we now add some sentences to lines 172-174 in section 2.3.1 and to the second comment, we modify the text in section 2.3.2.

*"• 2/ « The transfer coefficients ch and cm are sent by the atmosphere (Fig. 4). They are not calculated by the flux calculator. It is quite possible in reality to have stable air above the ice and unstable air above the open water. This yield very different transfert coefficients above the two surfaces. This should lead to very different flux compare to what the flux calculator computes. Note that if the calculations of these coefficients involve iterations in the atmosphere, this can indeed be complicated to compute them in the flux calculator. »*
*Referee # 1 wrote : « This (sic) two limitations should be explained to the reader, with if possible some descriptions of the potential impact, and technical constraints that explained the chosen algorithm. » and I don't see anywhere where this has been done.*
*Also the method does not separate scales. It treats an ice front separating an ice-covered area from an area of open water (large-scale heterogeneity) in the same way as ice fractured by leads (small-scale heterogeneity).*
*Please add something to address these remarks."*

We elaborate on the insensitivity of the transfer coefficients to the surface heterogeneity in the beginning and the end of section 2.3.2 (starting from line 204) in the revised manuscript.

*"3. Paragraphs lines 112-15 and lines 226-228 The sentence « The calculation of some fluxes cannot be reasonably implemented via bulk formulas that take only surface fields as input » sounds awkward to me, as does the sentence « do not only simply depend on surface fields » line 227 as noticed by Referee # 2.*
*Please consider rephrasing lines 112-113 more clearly with something like "However, the calculation of some fluxes does not depend only on surface fields and therefore are out of scope of the flux calculator capabilities. In particular, ..."*
*On line 114, consider changing "will still be calculated" with "are calculated".*
*Please consider rephrasing lines 226-228 as: "As stated above, the downward radiation fluxes ϕrad(x,y,t) (i.e. shortwave and longwave radiation) do not depend only on surface fields; they are thus entirely calculated by the atmospheric CCLM model and then passed through the flux calculator to the ocean model." "*

We change the mentioned lines as suggested in the revised manuscript.

*"4. Paragraph lines 239-242 The first and last sentences of this paragraph are redundant. Please consider merging them into "Note that all the presented formulas for fluxes might be changed, e.g. to involve different methods, within the flux calculator source code without further changing the model codes." Also, I think you should take the opportunity there to stress the fact that this is not true for the transfer coefficients ch and cm, which are calculated and sent by the atmosphere, see my remark 1. above."*

We change the mentioned lines as suggested and stressed the particular role of the transfer coefficients in the end of Sect. *2.3.2 Flux formulas* in the revised manuscript.

*"5. Paragraph lines 257-261 With your latest modifications, this paragraph does not read well. Please consider rephrasing for something like: "In the atmosphere, radiation fluxes are updated every hour; the timestep for the physics is 150s and is internally further subdivided to account for fast modes in the dynamics. The ocean model time step is 600s. The time step of the coupling between the two models is also set to 600s to temporally resolve strong wind gusts (Davis and Newstein, 1968). An investigation on the impact of different coupling time step sizes is planned for future work.""*

We change the mentioned lines as suggested in the revised manuscript.

*"6. Reference to the appendix For clarity, change the references to the appendices using the word "appendix" and not "Sect." and give the number of the appendix. For example, line 89, "Sect. B" should be "Appendix B" and at line 192 "Sect. D in the Appendix" should be "Appendix D2"."*

We change all cross references to the appendix using the word "appendix" in the revised manuscript.

*"7. Line 153 Consider changing « ... is employed, however, ... » for « ... is employed; however, ... »"*

We replace the comma by the semicolon in the revised manuscript.